# Inhibition allocates spikes during hippocampal ripples

Asako Noguchi[1,6], Roman Huszár[2,6], Shota Morikawa[1,3], György Buzsáki ⓘ [2,4✉] & Yuji Ikegaya ⓘ [1,3,5✉]

Sets of spikes emitted sequentially across neurons constitute fundamental pulse packets in neural information processing, including offline memory replay during hippocampal sharp-wave ripples (SWRs). The relative timing of neuronal spikes is fine-tuned in each spike sequence but can vary between different sequences. However, the microcircuitry mechanism that enables such flexible spike sequencing remains unexplored. We recorded the membrane potentials of multiple hippocampal CA1 pyramidal cells in mice and found that the neurons were transiently hyperpolarized prior to SWRs. The pre-SWR hyperpolarizations were spatiotemporally heterogeneous, and larger hyperpolarizations were associated with later spikes during SWRs. Intracellular blockade of $Cl^-$-mediated inhibition reduced pre-SWR hyperpolarizations and advanced spike times. Single-unit recordings also revealed that the pre-SWR firing rates of inhibitory interneurons predicted the SWR-relevant spike times of pyramidal cells. Thus, pre-SWR inhibitory activity determines the sequential spike times of pyramidal cells and diversifies the repertoire of sequence patterns.

[1] Graduate School of Pharmaceutical Sciences, The University of Tokyo, Tokyo 113-0033, Japan. [2] Center for Neural Science, New York University, 4 Washington Place, New York, NY 10003, USA. [3] Institute for AI and Beyond, The University of Tokyo, Tokyo 113-0033, Japan. [4] Neuroscience Institute, Department of Neurology, NYU Langone Medical Center and Center for Neural Science, New York, NY, USA. [5] Center for Information and Neural Networks, National Institute of Information and Communications Technology, Suita City, Osaka 565-0871, Japan. [6]These authors contributed equally: Asako Noguchi, Roman Huszár. ✉email: Gyorgy.Buzsaki@nyulangone.org; yuji@ikegaya.jp

The timing of neuronal spikes is critical for neural coding[1–3] and often generates spike sequences in which a set of neurons serially fire action potentials. Spike sequences are observed in many brain regions[4–7] and are believed to be involved in cognitive processes, including spatial memory[6–8] and decision-making[7].

Spike times during sequences are regulated at the millisecond level and are adaptively modifiable to generate diverse patterns of spike sequences. Spike sequences are exemplified by memory replays in sharp-wave ripples (SWRs), local field potentials that reflect transient excitatory drives, and high-frequency oscillations resulting from pyramidal cell-interneuron interactions in the hippocampus[9,10]. Hippocampal pyramidal cells that are sequentially activated during behavioral exploration are subsequently reactivated as a time-compressed sequence of spikes during SWRs while the animal is resting or sleeping[11]. This internal replay of behavioral experiences has been posited to contribute to memory consolidation, memory recall, and navigational planning[12,13]. Interestingly, spike sequences during SWRs can be replayed in both forward and backward directions depending on behavioral states[14,15]. Moreover, a neuron that participates in a sequence is reused in other sequences with different timings under different contexts[16,17]. This flexible recruitment of neuron assemblies may help increase the capacity of neural information using a limited number of neurons[18–21].

Several mechanisms have been proposed to explain sequential spiking during SWRs[16,22–24]. Pyramidal cells with higher spike rates fire earlier during SWRs than cells with lower spike rates[16], which indicates the contribution of the excitability of individual neurons to spike times during SWRs. Another, but not mutually exclusive, explanation attributes this phenomenon to anatomical and molecular factors, such as cell-to-cell differences in specific input connectivity and expression of receptors and channels[24]. However, mechanisms involving either fixed intrinsic properties or hard-wired neuronal networks cannot account for the flexible spike organization that can vary across SWR events.

Sequential spike activity can be locally generated in the CA1 subregion[23], and thus, the microcircuit-based dynamics within the CA1 subregion contribute to spike sequences. Interneurons are often embedded in spike sequences and may provide a temporal backbone for spiking pyramidal cells[23]. Consistent with this idea, local interneurons fire temporally coordinated spikes before or during SWRs[25,26]. In the present study, we recorded intracellular activity simultaneously from multiple CA1 pyramidal cells to examine the mechanisms underlying flexible sequential activity of CA1 pyramidal cells at the single-cell level and found that preceding GABAergic inhibition dynamically coordinates the spike times of individual pyramidal cells during SWRs.

## Results

**Simultaneous patch-clamp recordings from multiple hippocampal CA1 pyramidal cells in vivo.** We simultaneously patch-clamped multiple pyramidal cells in the ipsilateral CA1 region of the dorsal hippocampus of urethane-anesthetized mice while monitoring CA1 local field potentials (LFPs) (Fig. 1a). Recordings were excluded from the subsequent analyses if post hoc biocytin-based visualization, their recording sites, and their firing properties failed to identify them as CA1 pyramidal cells. As a result, we obtained two quadruple-patching datasets, 25 triple-patching datasets (Fig. 1b), 74 double-patching datasets, and 32 single-patching datasets from a total of 127 mice; the total number of analyzed cells was 64. Recording periods ranged from 1 min 53 s to 34 min 57 s (median = 8 min 43 s), during which a total of 5971 SWRs were recorded in LFPs.

Pooled data from all 49 pyramidal cells that emitted at least one spike during SWRs indicated that firing rates increased during SWRs (Fig. 1c bottom), as shown in previous studies using unit recordings[27]. The spike times were phase-locked to the ripple oscillations (Supplementary Fig. 1), as reported in freely moving animals[10]. Analyses of individual cells revealed that spike times were widely distributed around SWRs (Fig. 1c top; 32.7 ± 36.1 ms relative to the SWR onsets), indicating that individual neurons were capable of flexibly jittering their spikes relative to the SWR timing. Neither the means nor the standard deviations (SDs) of SWR-relevant spike times were correlated with the distances from the LFP electrode tips to the recorded cells (Supplementary Fig. 2), indicating that spike times were not influenced by cell locations, at least within our recording areas of $\varphi < 800\ \mu m$.

**Pre-SWR hyperpolarizations reflect inhibitory inputs.** Subthreshold membrane potentials ($Vm$) were averaged across all 5971 SWRs recorded from 64 cells. Consistent with a previous report[28], the averaged trace consisted of (i) slow depolarization starting ~1 s before SWRs, (ii) transient hyperpolarization ~50 ms before SWR onset, and (iii) large depolarization during SWRs (Fig. 2a, b top). The preceding hyperpolarization was reduced by intracellular application of 120 mM cesium fluoride and 1 mM 4,4′-diisothiocyanostilbene-2,2′-disulfonic acid (CsF-DIDS), which blocks GABA_A receptor-mediated Cl⁻ conductance[29] (Fig. 2b bottom, $n = 341$ SWRs in 31 cells). The $Vm$ changes immediately before SWRs (the $\Delta Vm_{pre}$ values) were $-0.20 \pm 1.8$ mV and $1.0 \pm 3.3$ mV for the control and CsF-DIDS conditions, respectively (Fig. 2b bottom inset, $P = 3.0 \times 10^{-29}$, $t_{6317} = -11.3$, Student's $t$-test, $n = 5971$ (control) and 348 (CsF-DIDS) SWRs). The more positive values of $\Delta Vm_{pre}$ in the presence of CsF-DIDS support that the main effect of CsF-DIDS was a reduction in inhibition. Indeed, inhibitory postsynaptic currents (IPSCs) were reduced under the voltage-clamp configuration using CsF-DIDS-loaded pipettes (Supplementary Fig. 3). We thus concluded that pre-SWR hyperpolarizations reflected GABAergic synaptic activity. Using a Cs⁺-based intrapipette solution, we voltage-clamped pyramidal cells at $Vm$ values of $-70$ and $10$ mV to isolate excitatory and inhibitory postsynaptic conductances (EPSGs and IPSGs), respectively. The mean EPSG and IPSG values during the period from 2000 ms before and 400 ms after the onset of SWRs were $3.9 \pm 3.1$ nS and $3.4 \pm 1.4$ nS, respectively. The mean traces of the time changes in EPSGs and IPSGs during 34 and 57 SWRs from 5 cells and their ratios are shown in Fig. 2c on the left. Similar traces were previously reported in awake mice[30]. We then plotted the time evolution of the mean conductances around SWR events in the EPSG-versus-IPSG space (Fig. 2c right) and found that IPSGs were dominant before SWRs and thereafter became linearly balanced with EPSGs. Compared to EPSGs, IPSGs during the pre-SWR periods were more variable in size across SWRs, which suggests that the SWR-to-SWR variability of $\Delta Vm_{pre}$ mainly reflected the variance in inhibition (Supplementary Fig. 4).

**Pre-SWR inhibition determines times of SWR-relevant spikes and depolarizations.** When GABA_A receptor-mediated Cl⁻ conductance was intracellularly blocked by CsF-DIDS, first spike times shifted to an earlier phase with respect to SWR onset (Fig. 3a, $P = 0.0064$, $D_{1185,230} = 0.12$, Kolmogorov–Smirnov test, $n = 1185$ and 230 spikes from 49 control cells and 31 CsF-DIDS-loaded cells, respectively). Thus, GABAergic inhibition delayed SWR-relevant spike times. We then focused on individual SWR events. The $Vm$ values of a CA1 pyramidal cell were often hyperpolarized immediately before SWRs, and the transient hyperpolarization varied in magnitude between SWR events

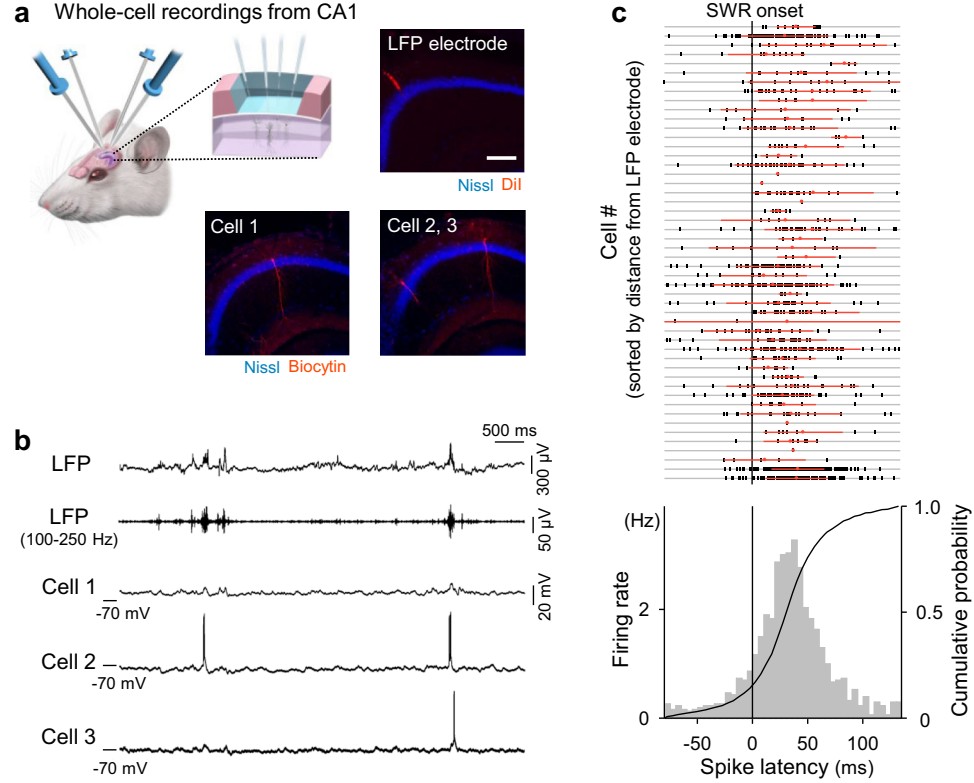

**Fig. 1 Simultaneous patch-clamp recordings from multiple hippocampal CA1 pyramidal cells in vivo. a** Schematic depiction of in vivo patch-clamp recordings from up to four neurons in the mouse hippocampus. The right photographs show post hoc biocytin-based identification of the recorded triplets and the LFP electrode track. Scale bar = 200 μm. 10 triplets were recorded independently. **b** Representative traces of LFPs, 100–250 Hz bandpass-filtered LFPs, and simultaneously recorded subthreshold $V$m values of three cells. **c** Top: All spike times in a total of 49 cells are shown in a raster plot, in which the cell order is sorted so that the distances from the LFP electrode tips are shorter in higher rows. The nine cases where we could not measure the distances are shown in the nine bottom rows. Each vertical tick mark indicates a single spike. Red dots and lines are the means ± SDs of spike times in the corresponding cells. Bottom: Distribution of all 1185 spike times in 49 cells relative to the SWR onsets. Source data is provided as a Source data file.

within a cell (Fig. 3b; mean ± SD: 0.49 ± 2.5 mV). Importantly, greater hyperpolarization was associated with later spikes relative to the SWR onset times (Fig. 3c, $R = -0.34$, $P = 0.0069$, $t$-test of the correlation coefficient, $n = 35$ spikes). When all 556 SWR-relevant spikes were pooled from 49 cells, the $\Delta V$m$_{pre}$ values were negatively correlated with the spike times (mean ± SD: 32.8 ± 27.7 ms) (Fig. 3d, $R = -0.32$, $P = 2.8 \times 10^{-14}$, $t$-test of the correlation coefficient).

Within each cell, the number of spikes per SWR event was positively correlated with $\Delta V$m$_{pre}$ (Supplementary Fig. 5a). The mean $\Delta V$m$_{pre}$ values in active cells that emitted at least one SWR-relevant spike during our recording period were not significantly different from those in silent cells that did not emit any spike during SWRs (Supplementary Fig. 5b), but the SDs of $\Delta V$m$_{pre}$ in active cells were higher than those of inactive cells (Supplementary Fig. 5c). These results suggest that presynaptic interneurons of active cells exhibit more different levels of activity across SWR events than those of inactive cells and that specific inhibition regulates the spikes of pyramidal cells during SWRs.

Even when CA1 neurons did not fire action potentials, they commonly depolarized during SWRs[28]. As in the case of spike times, the depolarization peak times varied across SWRs (Fig. 4a; mean ± SD: 43.7 ± 33.1 ms) and were negatively correlated with $\Delta V$m$_{pre}$ (−0.20 ± 1.8 mV) (Fig. 4b, c). The same result was reproduced in $V$m data obtained from unanesthetized head-fixed mice (Fig. 4d); the depolarization peak times and $\Delta V$m$_{pre}$ values were 37.5 ± 32.2 ms and −0.34 ± 2.0 mV, respectively. Thus, irrespective of whether neurons fired spikes, their subthreshold $V$m values were subject to inhibition tuning. Neither the means nor the SDs of the

depolarization peak times were correlated with the distances from the LFP electrode tips to the recorded cells (Supplementary Fig. 6).

To examine whether pre-SWR activity of interneurons causally contributes to the times of activity of pyramidal cells, we optogenetically inhibited parvalbumin (PV)-positive interneurons (Supplementary Fig. 7a). To express eNpHR, a natronomonas halorhodopsin, in PV-positive interneurons, the adeno-associated virus AAV-DIO-eNpHR-EYFP was injected into the dorsal hippocampal CA1 area of PV-Cre knockin mice. Green light stimulation (50-ms duration) was delivered every 3 s through an optic fiber inserted into a patch-clamp recording pipette. Under this stimulation protocol, some trials of optical stimulation happened immediately before SWR onset. In this case, we observed no apparent pre-SWR hyperpolarization in the patch-clamped pyramidal cells (Supplementary Fig. 7c). Indeed, larger $\Delta V$m$_{pre}$ values were observed when the light stimuli were timed closer to the SWR onsets (Supplementary Fig. 7d; $R = 0.39$, $P = 0.049$, $t$-test of the correlation coefficient, $n = 26$ SWRs from 10 neurons), indicating that inhibiting pre-SWR activity of PV-positive interneurons reduced pre-SWR hyperpolarization. Furthermore, the closer the light stimuli were timed to the SWR onset, the earlier the times of the depolarization peaks occurred (Supplementary Fig. 7e; $R = -0.41$, $P = 0.038$, $t$-test of the correlation coefficient, $n = 26$ SWRs from 10 neurons). Thus, the pre-SWR activity of PV-positive interneurons delays the timing of pyramidal neuron activity during SWRs.

**Pre-SWR hyperpolarizations are spatiotemporally dynamic.**
We conducted a pairwise analysis of subthreshold $V$m dynamics

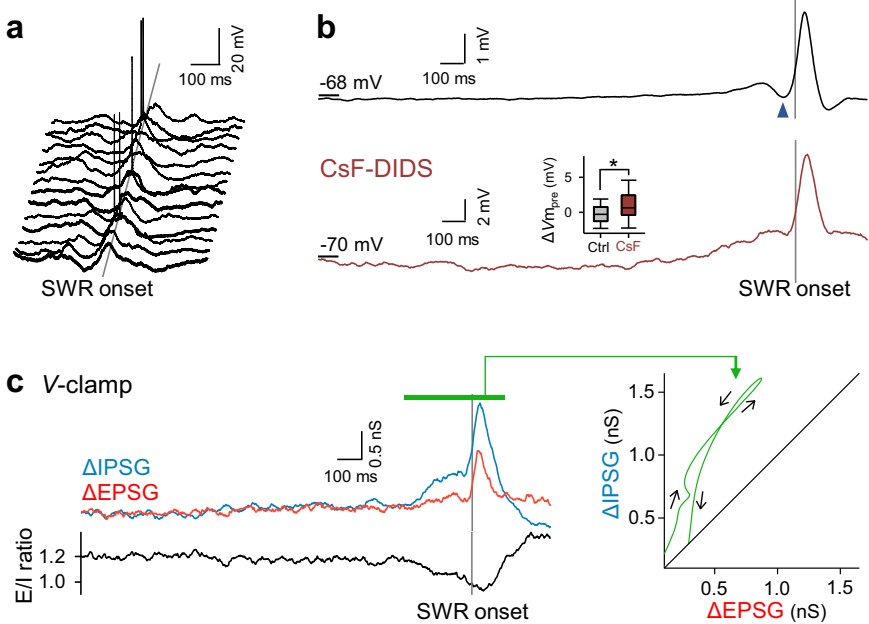

**Fig. 2 Pre-SWR hyperpolarizations reflect inhibitory inputs. a** Fourteen representative $Vm$ traces from 8 CA1 pyramidal cells, including noticeable pre-SWR hyperpolarizations, are aligned to the SWR onsets (gray straight line). **b** Top: A total of 5934 $Vm$ traces in 64 cells were averaged relative to the SWR onsets. The dark blue arrowhead indicates pre-SWR hyperpolarization. Bottom: The same as the top, but for 341 $Vm$ traces recorded in 31 CsF-DIDS-loaded cells. The pre-SWR hyperpolarization was reduced. The inset shows the quantification of $\Delta Vm_{pre}$ ($P = 3.1 \times 10^{-29}$, $t_{6317} = -11.3$, Student's $t$-test (two-sided), $n = 5971$ and 348 SWRs for control and CsF-DIDS conditions, respectively). Box plots indicate the median and 25–75% interquartile ranges; whiskers cover 10–90% quantiles. **c** Left: The top traces show SWR-aligned average of time changes in 34 EPSGs (red) and 57 IPSGs (blue) recorded from 5 cells. The bottom trace shows the ratio of the mean EPSGs to the mean IPSGs. Right: The temporal coevolution (black arrows) of EPSGs and IPSGs in the top left traces during SWRs (the period shown by the green bar) in the EPSG-vs.-IPSG space. IPSG increased earlier than EPSG. Source data is provided as a Source data file.

in a total of 46 double-patching datasets that contained at least 30 SWR events during the recording periods. The time differences in their depolarization peaks were distributed at ~0 ms (Supplementary Fig. 8; mean ± SD: 0.063 ± 41 ms). Therefore, neither cell tended to depolarize first during SWRs, and depolarization timing could be switched back and forth between the cells in a pair; note that in this analysis, the depolarization peak time of the lateral cell was subtracted from that of the medial cell.

We then extended the multicell analyses to 10 triple-patching datasets that contained at least 30 SWRs each. We found that within individual SWR events, different neurons received different magnitudes of inhibition (Fig. 5a top), indicating that inhibitory influences were spatially heterogeneous. For each SWR, the order of $\Delta Vm_{pre}$ among the three cells was plotted against the order of the depolarization peak times (Fig. 5a bottom). When the depolarization peak times were sorted to be $1 \to 2 \to 3$ across three cells, the $\Delta Vm_{pre}$ order $1 \to 2 \to 3$ was overrepresented, whereas the orders $3 \to 1 \to 2$ and $3 \to 2 \to 1$ were underrepresented, compared to the chance level estimated from 10,000 simulated trials in which $\Delta Vm_{pre}$s were randomly swapped across cells in each SWR (Fig. 5b). Therefore, even within a single SWR event, greater hyperpolarizations were associated with later depolarizations. We next analyzed sets of three simultaneously recorded cells that showed their depolarizations in the "forward" or "reverse" orders; note that the medial-to-lateral direction in the cell positions was defined herein as the forward order. More specifically, we sorted three cells in order from their medial to lateral positions, divided the order of depolarization times into two directions, i.e., from medial to lateral and from lateral to medial, and separately plotted the $\Delta Vm_{pre}$ values for these two cases (Fig. 5c). As a result, the orders of the $\Delta Vm_{pre}$ values were reversed when the orders of

depolarization times were reversed. These results suggest that identical triplets tend to receive the reverse order of inhibitory strengths when they exhibit depolarizations in the reverse order.

We next plotted the $\Delta Vm_{pre}$ of three cells along consecutive SWR events. In a representative triplet (Fig. 6a), a positive correlation of $\Delta Vm_{pre}$ was found between cells #1 and #3 but not between the other pairs of cells (mean $\Delta Vm_{pre}$ ± SD: 0.49 ± 2.1, −0.20 ± 1.4, 0.60 ± 3.2 mV for cells #1, 2, and 3, respectively); that is, greater hyperpolarization in cell #1 was associated with greater hyperpolarization in cell #3, and vice versa. Supplementary Fig. 9 summarizes the data of all 10 triplets, demonstrating that a fraction of the cell pairs displayed a significant $\Delta Vm_{pre}$ correlation. These partially correlated dynamics suggest the presence of cell assemblies[31,32]. Cell pairs showing stronger $\Delta Vm_{pre}$ correlations exhibited more strongly correlated depolarization peak times (Fig. 6b; $R = 0.44$, $P = 0.0020$, $t$-test of the correlation coefficients, $n = 46$ cell pairs). More strongly correlated $\Delta Vm_{pre}$s were found in cell pairs that were spatially closer together (Fig. 6c; $R = -0.41$, $P = 0.0052$, $t$-test of the correlation coefficients, $n = 46$ cell pairs). The mean correlation coefficients of $\Delta Vm_{pre}$ and the depolarization peak times were 0.33 ± 0.23 and 0.24 ± 0.19, respectively. The mean distances between the pairs of cells were 151 ± 149 µm.

**Pre-SWR interneuron firing predicts spike times of pyramidal cells during SWRs.** To further investigate the role of inhibition in shaping the spike times during SWRs, we recorded single-unit activity from 2490 putative pyramidal cells and 965 putative interneurons from the dorsal CA1 region of 6 freely moving mice. We first analyzed the firing rates of interneurons that were discharged at a fixed time with respect to SWRs. Consistent with a

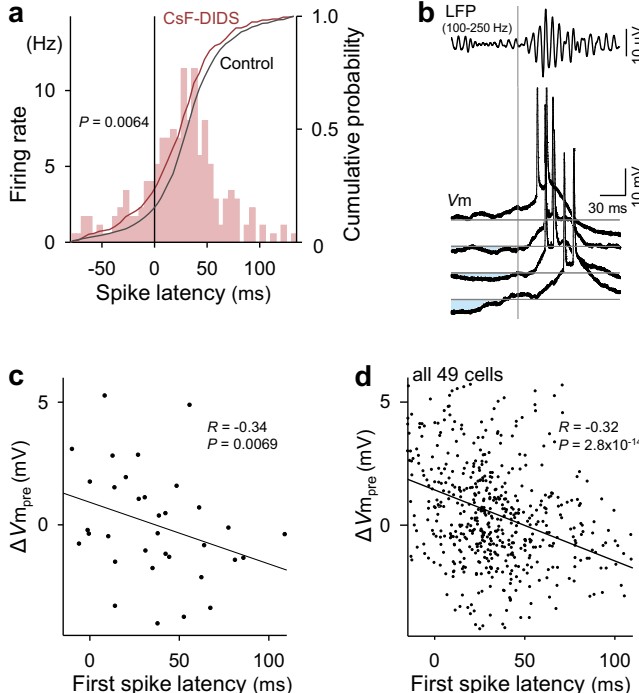

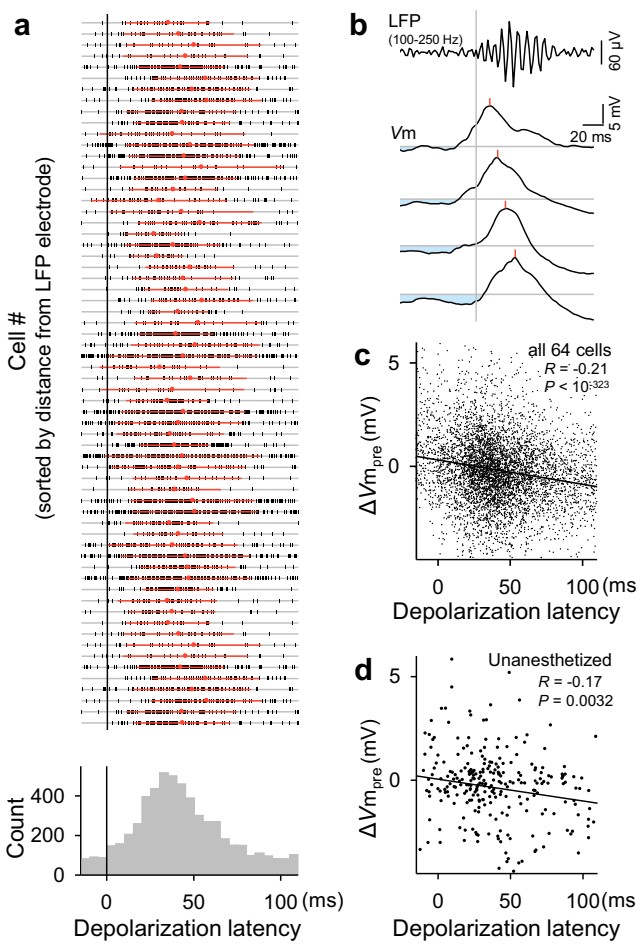

**Fig. 3 Pre-SWR inhibition determines SWR-relevant spike times. a** Distribution of 230 spike times in 31 cells relative to the SWR onsets for recordings under blockade of $Cl^-$ conductances with intracellular CsF-DIDS (colored bars and cumulative curve). The gray line shows the cumulative probability for recordings without CsF-DIDS. Significance was determined using the Kolmogorov–Smirnov test. **b** Four representative $Vm$ traces from a CA1 pyramidal cell, including action potentials during SWRs, are aligned to the SWR onsets (vertical gray line). The top trace shows the averaged bandpass-filtered LFPs. Larger prior hyperpolarizations (light blue areas) from the baseline (horizontal gray lines) are associated with later first spike times. **c** The relationship between $\Delta Vm_{pre}$ and the first spike times during SWRs. The line of best fit was determined using the least-squares method for a total of 35 spikes. Significance was determined using a $t$-test of the correlation coefficient (two-sided). **d** The same as (**c**), but for the pooled data of all 556 spikes from 49 cells. For visualization purposes, the range of the ordinate was limited, but all data points, including the outliers, were used for the linear regression. Source data is provided as a Source data file.

previous report[27], a subset of putative interneurons ($n = 75$; 7.78%) were discharged prior to the onset of SWRs (Fig. 7a). In each interneuron, the times of the firing rate increases were consistent among SWRs, although the magnitudes of the rate increases varied (Fig. 7b left). We divided the magnitudes of interneuron firing into quintiles (Fig. 7b left) to examine the effects of high *versus* low pre-SWR inhibition. The latencies to the first spikes of pyramidal cells were computed for each SWR, and their averages across SWRs belonging to the same quintiles were used to evaluate whether the spike times depended on the pre-SWR inhibition magnitudes (Fig. 7c). The difference in the mean spike latency between the first and fifth quintiles revealed the changes in spike times, which were assessed for significance by shuffling SWRs across quintiles (Fig. 7c inset). Of the 5639 paired pyramidal cells, 297 cells (5.3%) exhibited significantly delayed spike times following high pre-SWR inhibition (Fig. 7d left). The delays in the spike times of pyramidal cells were accompanied by reduced firing rates during SWRs (Fig. 7c, d right), indicating that our quintile-based assessment accurately extracted the effects of inhibition. For the activity of a single interneuron, the mean delay in the spike times of a pyramidal cell was ~6 ms, which translated to a delay of ~6% in its rank order in the sequence of pyramidal

**Fig. 4 Pre-SWR hyperpolarizations determine the times of SWR-relevant depolarizations. a** Top: All depolarization peak times in a total of 64 cells are shown in a raster plot, in which the cell order is sorted so that the distances from the LFP electrode tips are shorter in higher rows. The ten cases where we could not measure the distances are shown in the ten bottom rows. Each vertical tick mark indicates a single SWR event. Red dots and lines are the means ± SDs of depolarization peak times in the corresponding cells. Bottom: Distribution of the peak times of all 5682 depolarizations in 64 cells relative to the SWR onsets. **b** Four $Vm$ traces sampled from a representative CA1 pyramidal cell were aligned with the SWR onsets (vertical gray line). The top trace is the mean of the bandpass-filtered LFPs. Light blue areas indicate preceding hyperpolarizations from the baseline (horizontal gray lines). Red lines indicate the depolarization peak times during SWRs. **c** The relationship between $\Delta Vm_{pre}$ and the depolarization peak times during SWRs. The line of best fit was determined using the least-squares method for a total of 5971 SWRs in all 64 cells. Significance was determined using a two-sided $t$-test of the correlation coefficient. For visualization purposes, the range of the ordinate was limited, but all data points, including the outsiders, were used for the linear regression. Larger preceding hyperpolarizations from the baseline are associated with later depolarization times. **d** The same as (**c**) but for 285 SWRs recorded from all 9 cells in unanesthetized head-fixed mice. Source data is provided as a Source data file.

cell firing (Fig. 7e). Given that spike sequences during longer-duration SWRs include cells with lower firing rates and longer spike latencies[16], it is possible that the durations of SWRs mediated the relationship between pre-SWR firing rates of interneurons and spike latencies of pyramidal cells. Therefore, we quantified the durations of SWRs for each quintile of pre-SWR firing rates of interneurons (Supplementary Fig. 10). Increased

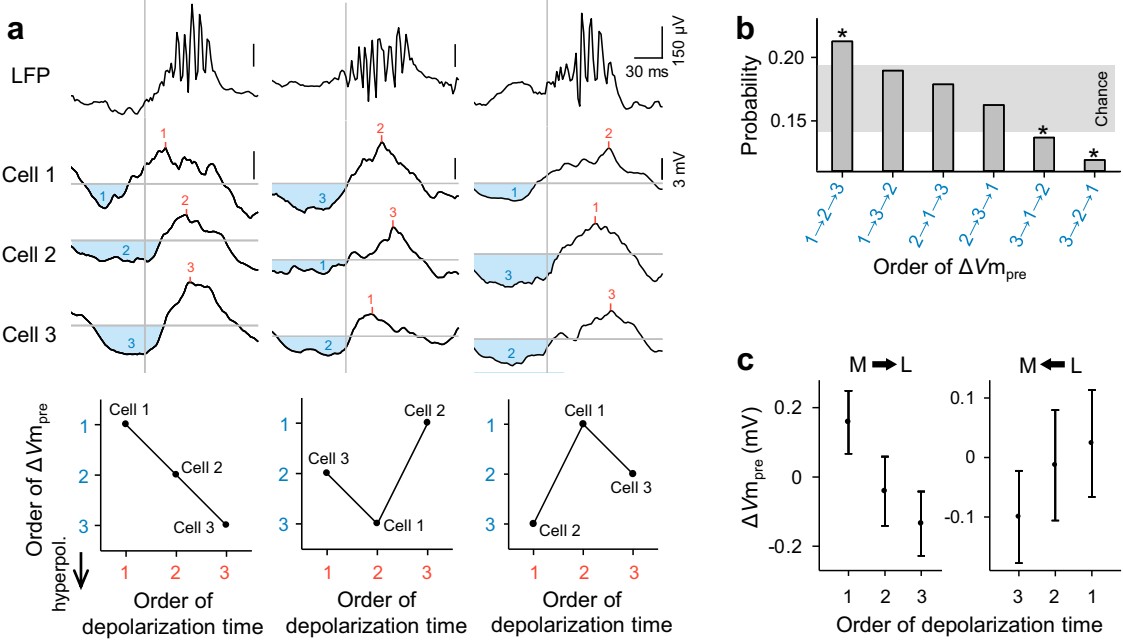

**Fig. 5 Pre-SWR hyperpolarizations in hippocampal CA1 pyramidal cells are associated with SWR-relevant depolarization times among multiple pyramidal cells. a** Representative $Vm$ traces sampled from three simultaneously recorded CA1 pyramidal cells (cells #1, #2, and #3) during three SWR events. The top traces are raw LFPs. Light blue areas indicate hyperpolarizations preceding SWRs, whereas red lines indicate the depolarization peak times during SWRs. Each bottom plot indicates the rank relationship between the orders of $\Delta Vm_{pre}$ and the depolarization peak times in 3 cells during the corresponding SWR. The orders correspond to the numbers shown in the traces (blue: $\Delta Vm_{pre}$, red: depolarization peak time). **b** Distribution of the orders of $\Delta Vm_{pre}$ under the conditions that the depolarization peak times across three cells were sorted as $1 \rightarrow 2 \rightarrow 3$. *$P < 0.05$ (two-sided) was determined using a random sampling method with 10,000 simulated trials in which the $\Delta Vm_{pre}$ values were randomly exchanged across cells within each SWR. The gray zone indicates the 95% confidence interval estimated from the simulated trials. **c** Among three simultaneously recorded cells, cells that showed larger $\Delta Vm_{pre}$s depolarized later, whereas the same cells depolarized earlier when they showed smaller $\Delta Vm_{pre}$s. The orders of cells were sorted depending on the cell locations, from the medial to lateral direction (left, herein "forward") and from the lateral to medial direction (right, herein "backward"). Data are shown as the mean ± SEM of 333 and 335 triplets for the forward and backward directions, respectively. Source data is provided as a Source data file.

pre-SWR firing rates of interneurons were not associated with changes in the SWR durations, indicating that later spike latencies after higher pre-SWR firing rates of interneurons were not merely a result of longer SWR durations.

The effect of pre-SWR inhibition on the structure of sequences was further investigated by fitting a hidden Markov model (HMM) to pyramidal cell firing during SWRs belonging to the first and fifth quintiles (Fig. 7f)[33]. Sequences in a given quintile were explained less effectively by HMMs fitted to the other quintile than by control HMM fits where SWRs were shuffled across quintiles (Fig. 7g left, $n = 128$, $P = 0.018$, one-tailed Student's $t$-test). We performed 100 independent shuffles and consistently reproduced the same effect (Fig. 7g right). These results again suggest that the pre-SWR firing rates of a given interneuron modulated spike sequences of pyramidal cells during SWRs.

We next examined whether the relationship between the activity of pre-SWR firing interneurons and the spike latencies of pyramidal neurons was also observed in artificially imposed network patterns (Fig. 8). For this purpose, we sparsely expressed channelrhodopsin-(ChR2)-enhanced yellow fluorescent protein (EYFP) fusion protein in a subset of CA1 pyramidal cells and optogenetically induced SWR-like high-frequency oscillations[10,23] by delivering 125-to-145-ms blue light pulses at intervals randomly chosen from a range between 1.3 and 1.7 s. The properties of the artificial SWRs did not depend on the blue light intensities (Supplementary Fig. 11). For analyses, we focused on a total of 38 interneuron-pyramidal cell pairs, in which pre-SWR interneuron firings modulated pyramidal spike times in spontaneously occurring SWRs (Fig. 7), and the pyramidal neuron

discharged more than 10 spikes across light pulses. Light stimulation-induced changes in the firing rates were highly variable across the interneurons (Fig. 8b), presumably due to the sparse expression of ChR2 and the anatomical structure of the local circuitry. This variability indicates that we could artificially activate or suppress some of the interneurons that were activated before SWR onsets in the spontaneous conditions. We then took advantage of this artificially induced increase or decrease in the firing rates of the pre-SWR activated interneurons and examined whether the induced changes in the interneuron firing rates resulted in the corresponding changes in the spike times of the associated pyramidal cells. We found a significant relationship between the within-pulse interneuronal firing rates and the spike latencies of pyramidal cells even under these artificial conditions (Fig. 8c; Spearman $\rho = 0.365$, $P = 0.024$, $n = 38$ pairs). Moreover, pre-SWR interneurons were activated earlier than the corresponding pyramidal cells on average (Fig. 8d; $n = 32/38$ pairs, 84.2%). These results further support the idea that variable interneuronal activity affected the spike latencies of their corresponding pyramidal cells.

**Heterogeneity in SWR-relevant interneuronal activity.** In the analyses above, we propose that heterogeneity in interneuronal firing affects spike sequences of CA1 pyramidal cells. However, SWR-relevant activity of interneurons is known to be highly synchronous (Fig. 9a)[34], and may be unable to regulate the spike times of pyramidal cells at such a fine timescale. However, we also recognized further heterogeneity underlying this synchrony (Fig. 9b). Specifically, in different SWR subsets, interneurons were more or less synchronous than they were on average. To quantify this observation, we fit a

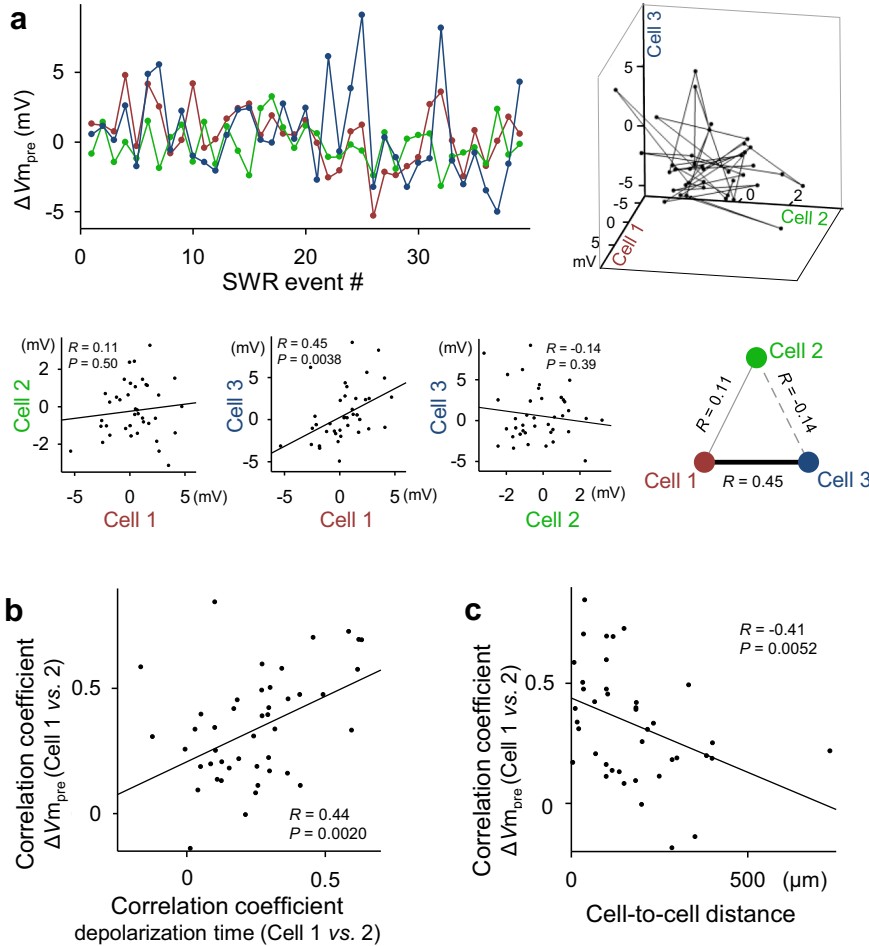

**Fig. 6 Pre-SWR hyperpolarizations are spatiotemporally dynamic. a** Top: Representative plot of $\Delta Vm_{pre}$ for three simultaneously recorded cells along a time series of SWR events (left) and its three-dimensional visualization (right). Each dot in the graph indicates $\Delta Vm_{pre}$ for a single SWR event. Bottom: Correlations of $\Delta Vm_{pre}$ between all three possible pairs of cells #1, #2, and #3. All the remaining data are shown in Supplementary Fig. 9. **b** The correlation coefficients of $\Delta Vm_{pre}$ are plotted against those of the depolarization peak times for all 46 cell pairs. **c** The correlation coefficients of $\Delta Vm_{pre}$ are plotted against the distances between the cell pairs. In all correlation graphs, the lines of best fit were determined using the least-squares method and statistically evaluated using a $t$-test of the correlation coefficient (two-sided). Source data is provided as a Source data file.

statistical model (generalized mixture model with multivariate Poisson observations) to extract ensembles of the SWR-relevant interneuron firing (Fig. 9c–e)[35]. Across all the relevant sessions that had interneurons firing prior to SWRs (as we analyzed in Fig. 7), there was further heterogeneity in interneuron firing throughout SWRs. More than 2 interneuron ensembles could be detected across all sessions (Fig. 9f), which reflected more structure than could be expected from different average SWR-related firing rates (Fig. 9g). The ensembles reflected co-fluctuations around the interneurons' average firing rates (Fig. 9h), indicating that interneurons fired reliably and to different extent across ensembles, rather than participating exclusively in specific SWR subsets. These results suggest that downstream pyramidal cells would receive different amounts of inhibition across different SWRs, which further supports the idea that heterogeneous inhibitory control of pyramidal cells could promote a diverse repertoire of sequential patterns.

## Discussion

We discovered that $\Delta Vm_{pre}$ in hippocampal CA1 pyramidal cells was negatively correlated with the spike times during SWRs and that the pre-SWR firing rates of interneurons partially predicted the spike times of pyramidal cells. These findings propose a novel role of inhibition in SWRs; that is, pre-SWR inhibition coordinates spike sequences during SWRs, whereas during-SWR and

post-SWR inhibition was previously reported to suppress the activity of competing cell assemblies and increase firing specificity[23,36]. Our findings are also consistent with in vitro studies demonstrating that artificial stimulation of a single interneuron can initiate SWRs[37] and sequential spikes of pyramidal cells in hippocampal slices[38].

We found that 5.3% of the interneuron-pyramidal cell pairs had significantly negative correlations between the pre-SWR firing rates of the interneurons and the spike latencies of the pyramidal cells. This small proportion may be accounted for by the fact that unit recordings do not identify directly connected interneuron-pyramidal cell pairs, whereas whole-cell recordings capture the net inhibitory input into the recorded cells. Moreover, inhibition does not directly lead to firing, and its contribution is hardly detected using unit recordings and can be investigated more appropriately using whole-cell recordings. In addition, we consider that pre-SWR inhibition is one of the factors that can regulate the spike times of pyramidal cells. Other factors, such as dynamic excitation[16,28], also contribute to spike times, making it difficult to isolate the pure effect of pre-SWR inhibition. For example, cells that fire earlier in spike sequences exhibit higher firing rates, which suggests a functional link between excitability and spike times[16].

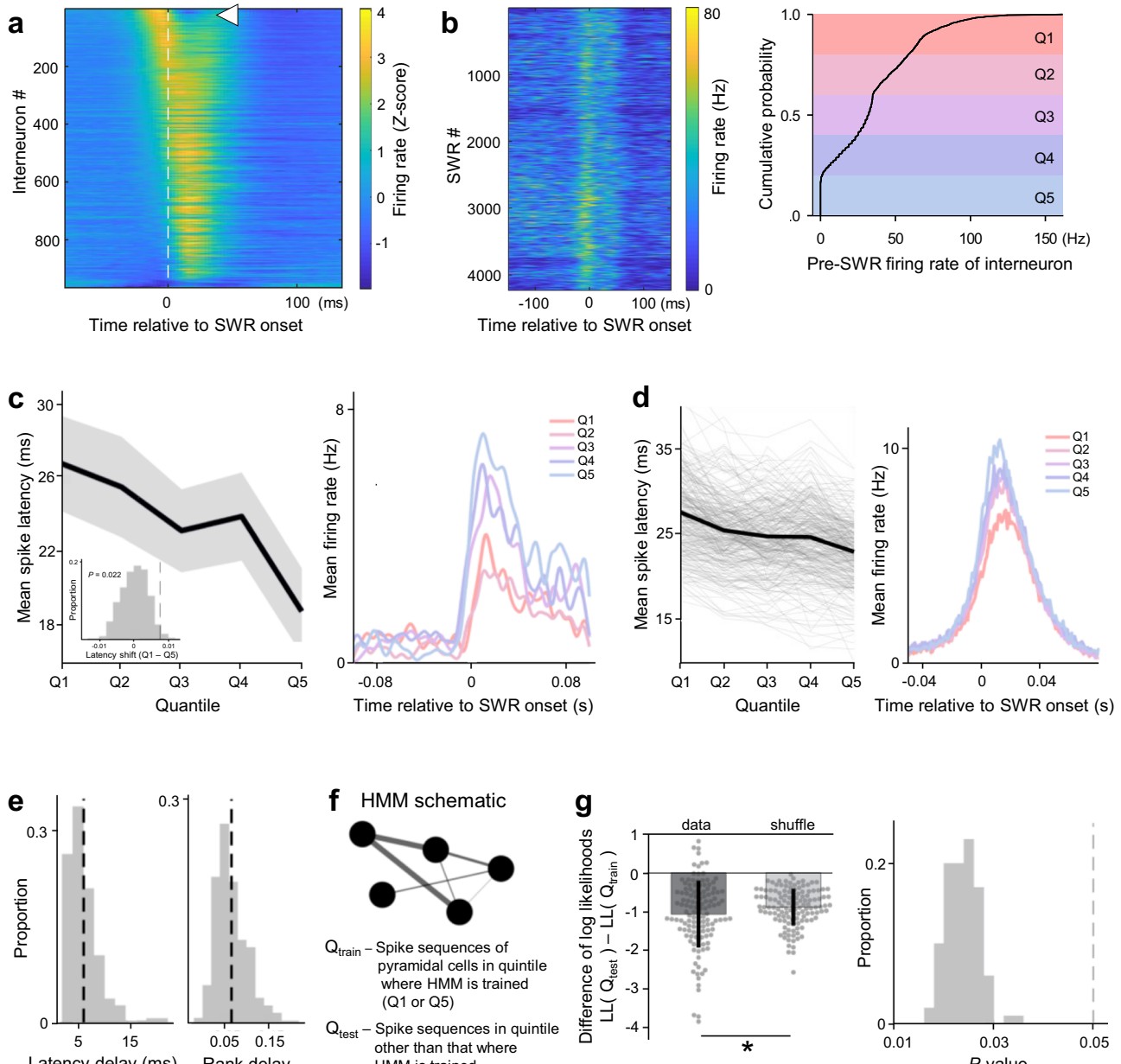

**Fig. 7 Pre-SWR interneuron firing predicts spike times of pyramidal cells during SWRs. a** Average interneuron firing rates (*Z*-score) around SWR onsets, ordered by average rate 30 ms prior to SWR onset. The white arrowhead highlights the interneuron shown in (**b**). **b** Left: Consistent rate increases before SWRs despite heterogeneous firing rate magnitudes in an example interneuron. Right: Spike trains were smoothed (15 ms FWHM Gaussian), and the cumulative distribution of pre-SWR firing rates (computed in 30-ms windows before SWR onset) was divided into quintiles (Q1–Q5). **c** Left: Mean spike latencies (±SEM) in an example pyramidal cell as a function of quintiles Q1–Q5. Inset: To assess statistical significance, the difference in mean spike latency across Q1 and Q5 (7.6 ms for this example) was compared against a null distribution of spike latency differences obtained by shuffling SWRs across quintiles (*n* = 500). Right: Average firing rates across quintiles Q1–Q5 reveal that spike latency delay was associated with a decrease in firing rate. **d** Same as (**c**) but for *n* = 297 pyramidal cells whose spike latencies were significantly delayed. **e** Histograms of average latency (left) and rank (right) changes of all pyramidal cells that delayed their firing depending on the quintiles. Average latency delay of ~6 ms was associated with a rank-order change of ~6.6%. **f** Schematic of an HMM for quantifying sequential firing within SWRs. Circles denote patterns of pyramidal cell co-firing and edges of different widths represent transition probabilities between patterns. HMMs were fitted to pyramidal firing in all SWRs associated with a particular quintile (e.g., Q1; $Q_{train}$), and the likelihood of pyramidal firing in SWRs of the other quintile (e.g., Q5; $Q_{test}$) was assessed under this model. **g** Left: The difference in log-likelihoods between the test and training quintiles quantifies the HMM goodness-of-fit to sequential firing in SWRs in the held-out test quintile. Significance was assessed against a null distribution in which SWRs were shuffled across quintiles (*n* = 128 quintile differences; *P* = 0.018, one-sided). The data are presented as means ± SDs. Right: *n* = 100 independent shuffles were performed to control for spurious resampling effects. The resulting *P*-values are shown as a histogram.

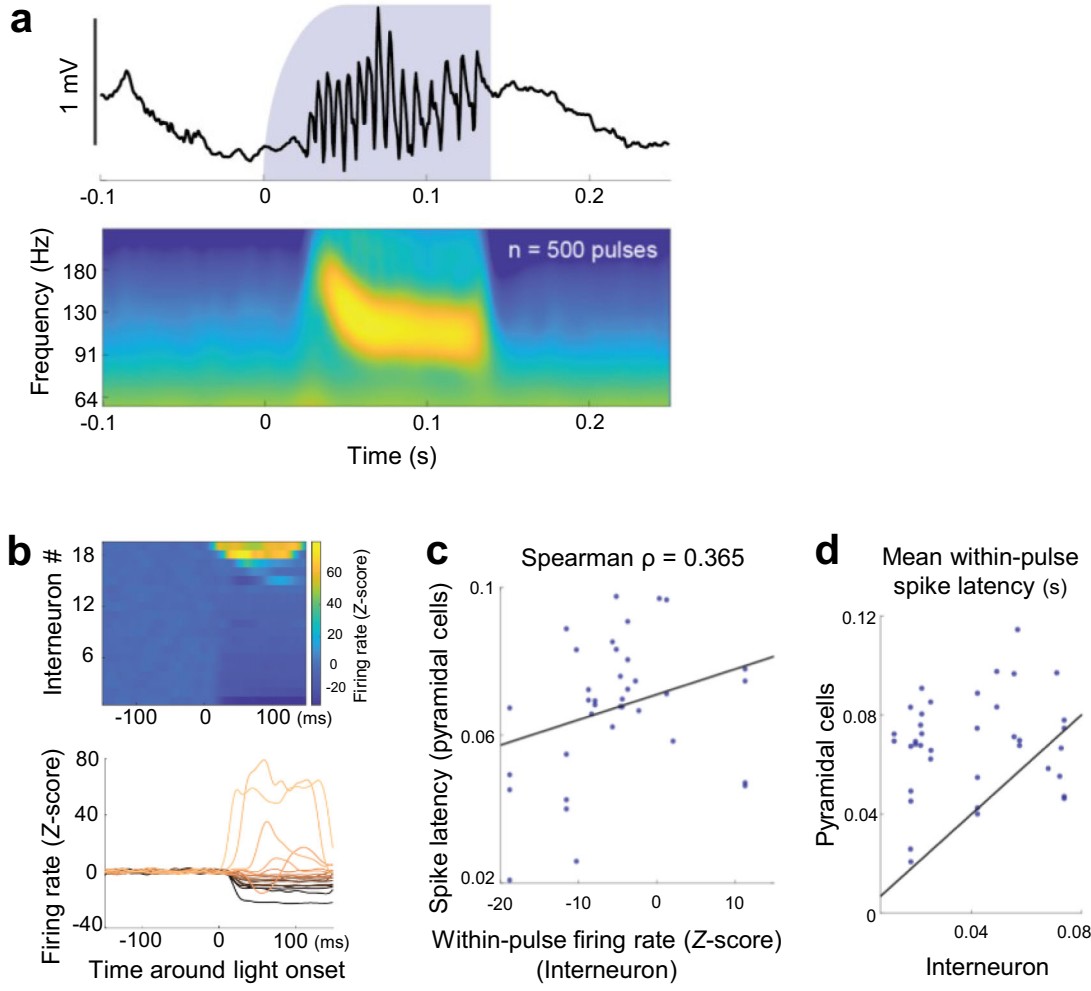

**Fig. 8 Interneuron firing predicts the spike latency of pyramidal cells during optogenetically induced SWRs. a** Top: CA1 local field potential (LFP) around an example of optogenetically induced SWR-like high-frequency oscillations. Bottom: Light onset-triggered average of LFP wavelet spectrogram of 500 optogenetically induced SWRs in an example recording session. **b** Average peristimulus time histogram of all interneurons ($n = 19$) that both displayed pre-SWR firing and were associated with significantly delayed spike timing of at least one pyramidal neuron (defined in Fig. 7). The values are Z-scored based on the 150-ms baseline period prior to the light onset followed by subtraction of the mean value. **c** The spike latencies of pyramidal neurons, averaged across light pulses, as a function of the within-pulse firing rates of the corresponding interneurons. Only interneuron-pyramidal pairs in which an effect of pre-SWR inhibition on pyramidal spike timing was observed were considered. The black line represents the least-squares regression line. Spearman $\rho = 0.365$, $P = 0.024$, a $t$-test of the correlation coefficients (two-sided), $n = 38$ pairs. **d** Average light pulse-associated spike latencies of the interneuron and pyramidal cell in each pair. Black, diagonal showing equal spike latencies. In 32/38 pairs (84.2%), the interneuron spike latencies were shorter than those of the associated pyramidal cells.

Consistent with a previous study in which subthreshold $Vm$ activity of single neurons was recorded during in vivo SWRs[28], we observed approximately three phases in the $Vm$ dynamics around SWRs: (i) pre-SWR ramps, (ii) sharp depolarizations during SWR, and (iii) post-SWR hyperpolarizations. In addition, we found that reliable, but varying-magnitude, hyperpolarizations occurred immediately before SWRs and demonstrated a crucial role in spike sequences. A previous study also demonstrated that pyramidal cells with more depolarized $Vm$ showed earlier depolarization peak times[28]; however, in this analysis, SWR-relevant $Vm$ dynamics were pooled from all SWRs in all cells recorded, and the variability across SWRs or cells was not considered. In the present study, we directly compared pre-SWR inhibition to the subsequent spike time of a pyramidal cell in each SWR event and extended this approach to multiple whole-cell recordings and provided evidence that pre-SWR inhibition is associated with sequential activity among multiple pyramidal cells.

We showed no significant relationships between the relative times of the cell activity to the SWR onsets and the distances between the recorded cells and the LFP electrodes. A previous study reported that SWRs propagate at 300–400 μm/ms along the septotemporal axis of the hippocampus[39]. The range of the recording area in the present study was $\varphi < 800$ μm. Thus, the time lag of SWRs along the positions of LFP electrodes is estimated to be <2 ms, which is sufficiently small for the time scales in our analyses (tens of milliseconds) and does not affect our results.

Neurons are flexibly recruited in different orders for various sequences, and sequences can be played either forward or backward[14,15]. Because our data were recorded mainly from head-fixed, anesthetized mice, we could not determine forward or reverse replays of behaviorally relevant spike sequences. However, at least in terms of the flexible sequence compositions of identical groups of cells, pre-SWR inhibition with degrees of freedom could be a candidate mechanism rather than neural mechanisms based on hard-wired microcircuitry. In accordance with this idea,

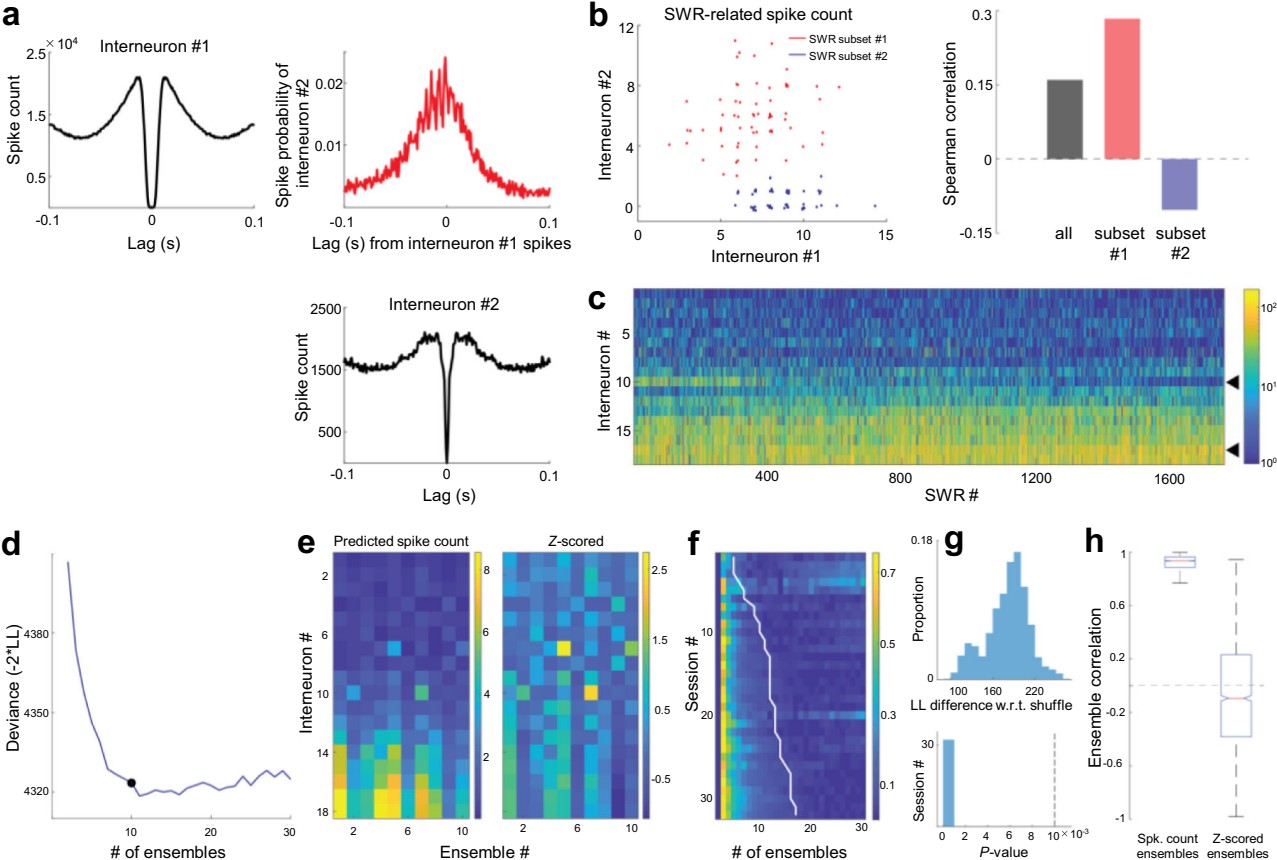

**Fig. 9 Heterogeneity in SWR-related interneuronal activity. a** Auto- and cross-correlograms of two example interneurons computed from spikes emitted during SWRs. Note the highly rhythmic spike relationship throughout SWRs. **b** Left: Spike counts of example interneurons (shown in (**a**)) in two subsets of SWRs (red, blue) recorded in the same session. Each dot indicates a single SWR. Right: SWR-related spike count correlations in all SWRs (black) and the two subsets (red, blue) highlighted on the left. **c** SWR-related spike counts of all interneurons recorded in an example session, sorted according to mean firing rate. Black triangles highlight interneurons shown in (**a**) and (**b**). **d** The spike count matrix in (**c**) was modeled using a generalized mixture model with Poisson observations ("ensembles"). The optimal number of ensembles was identified as the point of maximum curvature in the cross-validated deviance curve ($-2 \times$ log-likelihood). **e** Left: Model-predicted interneuron spike counts in each ensemble. *Right*: Spike counts are expressed as Z-scores with reference to spike count distributions across all SWRs. **f** Deviance curves (rows, normalized between 0 and 1) in all sessions used for the analysis in Fig. 7g. The white line denotes the optimal number of interneuron ensembles. **g** Top: For each selected model, log-likelihood on the test set was compared against a surrogate dataset in which spike counts were shuffled across SWRs for each individual interneuron (see "Methods"), thereby yielding a P-value (bottom; $n = 32$ sessions, two-sided, one-sample $t$-test). Dashed vertical line indicates $P = 0.01$. **h** Ensemble-ensemble correlations ($n = 2197$ pairs) when the contributions of interneurons were expressed as predicted spike counts (left) or Z-scored (right). The central mark and edges indicate the median and the 25th/75th percentiles, respectively, while the whiskers extend to the most extreme data points.

Fig. 5c suggests that the identical cell sets could receive inhibition in the reverse order and exhibit depolarizations in the reverse order. On the other hand, the flexibility of inhibition seems inconsistent with the strict forward and reverse replays. The behaviorally relevant excitatory drives from the CA3 and the entorhinal cortex, and the partial contribution of the anatomical structure may confine the order to activation.

It is intriguing that subthreshold $V$m formed sequences among multiple pyramidal cells. Sequences have been referred to as the firing order, but our data suggest that even cells that are silent in a sequence are ready to participate in the sequence. Subthreshold dynamics may represent preadaptive states that are beneficial to rapid and flexible plasticity in learning and memory or may represent latent traces of plasticity that reflect past neural history. Consistent with this idea, a recent study has demonstrated that optogenetic prolongation of SWRs recruits spikes from a low-firing population of pyramidal cells and extends the sequences[16], suggesting that some form of sequential activity pre-exists in subthreshold $V$m dynamics.

As a factor in the similar dynamics of the $\Delta V$m$_{pre}$ values in cell pairs, we showed that spatially clustered neurons receive more similar pre-SWR inhibition, which is in line with a distance-dependent decrease in the connection probability and strength of GABAergic synapses. While the probability of place field representation is thought to be randomly distributed in CA1 pyramidal cells, regardless of their relative positions[40], our result leads to the question that spatially closer pyramidal neuron pairs should behave similarly to more distant pairs. Additional studies with large-scale single-unit recordings are needed to address this issue.

The currently dominant view about CA1 neural plasticity assumes pyramidal-pyramidal neuronal interactions driven from the upstream regions (often from CA3 pyramidal cells). Our data suggest the role of inhibitory drives in spike sequences of pyramidal cells, consistent with observations that synaptic connections between pyramidal cells and interneurons local to CA1 are also plastic[23,41–45]. In addition to the non-uniform innervation of interneurons by pyramidal cells, the mechanisms underlying the heterogeneous inhibition of cell assemblies or their sequential

activity may involve plastic changes in synaptic transmissions between pyramidal cells and interneurons[45].

It remains unclear how interneurons can be activated before SWRs. Both CA1 pyramidal cells and interneurons receive excitatory inputs from the CA3 network, a major site of SWR initiation[9]. One possibility is that feedforward inhibition is more strongly convergent than feedforward excitation. Excitatory synapses on interneurons usually have higher transmission efficiency than those on pyramidal cells[46,47]; thus, interneurons can be activated earlier than pyramidal cells[48]. Indeed, CA1 pyramidal cells and interneurons respond differently to CA3 network synchrony in a nonlinear manner[49], and a subset of interneurons are rapidly recruited in the early phase of SWRs[27]. It is also intriguing to consider the involvement of the CA2 region. CA2 neurons may directly activate CA1 interneurons while triggering SWRs in the CA3 region, producing a multisynaptic delay of excitation[50]. Consistent with the possible inhibitory role of CA2 pyramidal cells, their chemogenetic silencing facilitates reactivation of CA1 cell assemblies while it reduces the temporal precision of the reactivation, often advancing the reactivation times[51].

Diverse classes of interneurons exist in the hippocampus, exhibiting different patterns of SWR-related spikes[52]. Parvalbumin-expressing basket and bistratified cells fire spikes that were phase-locked to SWRs, whereas parvalbumin-expressing axo-axonic cells are more likely inhibited during SWRs[52]. Pre-SWR inhibitory synaptic inputs may be produced by parvalbumin-expressing basket and bistratified cells, which are directly innervated by CA3 pyramidal cells and can discharge tens of milliseconds before SWRs. In support of this idea, some types of interneurons in the CA1 pyramidal cell layer exhibit peaks in their firing rates both before and after the maximum discharge probability of CA1 pyramidal cells, suggesting that these cells are activated by CA3 input before the discharge of CA1 pyramidal cells and are inhibited in the middle of SWRs[27]. However, a recent study demonstrated more detailed heterogeneity of interneurons[26]. Further investigations using cell type-specific observations and manipulations of the activity of interneurons are necessary to clarify the interneuron class responsible for spike sequencing.

## Methods

**Data.** All data analyzed in this study were newly collected for this study and were not reanalyses of past data.

**Animals.** Animal experiments for patch-clamp recordings were performed with the approval of the Animal Experiment Ethics Committee at The University of Tokyo (approval number: P29-9) and according to the University of Tokyo guidelines for the care and use of laboratory animals. These experimental protocols were carried out in accordance with the Fundamental Guidelines for Proper Conduct of Animal Experiment and Related Activities in Academic Research Institutions (Ministry of Education, Culture, Sports, Science and Technology, Notice No. 71 of 2006), the Standards for Breeding and Housing of and Pain Alleviation for Experimental Animals (Ministry of the Environment, Notice No. 88 of 2006) and the Guidelines on the Method of Animal Disposal (Prime Minister's Office, Notice No. 40 of 1995). All animals were housed under a 12/12-h light-dark cycle (light from 07:00 to 19:00) at $22 \pm 1\,°C$ with food and water provided ad libitum. For chronic extracellular recordings, all experiments were conducted with the approval of the Institutional Animal Care and Use Committee of New York University Medical Center.

**Patch-clamp and LFP recording.** Whole-cell recordings were obtained from postnatal 28- to 40-day-old male ICR mice (Japan SLC, Shizuoka, Japan)[53,54]. After exposure to an enriched environment for 30 min, the mice were intraperitoneally anesthetized with 2.25 g/kg urethane. Anesthesia was confirmed by the absence of paw withdrawal, whisker movement, and eyeblink reflexes. The skin was subsequently removed from the head, and a metal head-holding plate was fixed to the skull. A craniotomy of $2.5 \times 2.0\,mm^2$ was performed[55–57]. The exposed hippocampal window was covered with 1.7% agar at a thickness of 1.5 mm. For recordings from unanesthetized mice, mice were implanted with metal head-holding plates under short-term anesthesia with 2–3% isoflurane. After more than

24 h of recovery, the mice received head-fixation training on a custom-made stereotaxic fixture for 1–2 h per day. The training continued for up to 5 days until the mice learned to remain calm. To increase the occurrence of SWRs, the mice were exposed to an enriched environment for 1–2 h before electrophysiological experiments[58]. One LFP-recording electrode and four patch-clamp pipettes were serially inserted into the hippocampus. LFPs were recorded from the dorsal CA1 region using a tungsten electrode (3.5–4.5 MΩ, catalog #UEWMGCSEKNNM, FHC, USA) coated with a crystalline powder of 1,1′-dioctadecyl-3,3,3′,3′-tetramethylindocarbocyanine perchlorate (DiI). Whole-cell patch-clamp recordings were obtained from neurons in the CA1 pyramidal cell layer (AP: −1.0 to −3.0 mm; ML: 1.0–2.5 mm; DV: 1.1–1.3 mm) using borosilicate glass electrodes (3–8 MΩ). Pyramidal cells were identified based on regular spiking properties in response to step-pulse current injection and morphological features, including apical, oblique, and basal dendrites with spines, in post hoc histology. A cell was discarded unless it was identified as a pyramidal cell. For current-clamp recordings, the intrapipette solution consisted of the following reagents: 120 mM K-gluconate, 10 mM KCl, 10 mM HEPES, 10 mM creatine phosphate, 4 mM MgATP, 0.3 mM Na₂GTP, 0.2 mM EGTA (pH 7.3), and 0.2% biocytin. The intrapipette solution to prevent Cl⁻-mediated inhibitory currents consisted of the following reagents: 120 mM CsF, 10 mM KCl, 10 mM HEPES, 5 mM EGTA, and 1 mM DIDS. Liquid junctions were corrected offline. Cells were discarded when the mean liquid resting potential exceeded −50 mV and the action potentials were below −20 mV. For voltage-clamp recordings, the intrapipette solution consisted of the following reagents: 130 mM $CsMeSO_4$, 10 mM CsCl, 10 mM HEPES, 10 mM phosphocreatine, 4 mM MgATP, 0.3 mM NaGTP, and 10 mM QX-314. Cells were discarded if the access resistance exceeded 60 MΩ. Signals recorded by LFP electrodes were amplified using a DAM80 AC differential amplifier. Signals recorded by patch-clamp electrodes were amplified using MultiClamp 700B amplifiers. Both types of signals were digitized at a sampling rate of 20 kHz using a Digidata 1440A digitizer that was controlled by pCLAMP 10.3 software (Molecular Devices).

**Histology.** Following each experiment, the electrode was carefully withdrawn. The mice were transcardially perfused with 4% paraformaldehyde followed by overnight postfixation. The brains were sagittally sectioned at a thickness of 100 μm using a vibratome. The sections were incubated with 2 μg/ml streptavidin-Alexa Fluor 594 (647 for the experiments using PV-Cre mice in Supplementary Fig. 7) conjugate and 0.2% Triton X-100 for 4 h, followed by incubation with 0.4% NeuroTrace 435/455 Blue Fluorescent Nissl Stain (Thermo Fisher Scientific; N21479) for 2–4 h. For each section, 6–31 fluorescent images were acquired at a Z-step size of 2.0 μm using an Olympus FV1000 or FV1200 confocal microscope with a ×10 dry objective lens (numerical aperture: 0.4) and were stacked as maximum-intensity Z-projections. The location of each LFP electrode was detected based on the track of DiI fluorescence. The cell morphology was evaluated based on the silhouette of Alexa Fluor 594 fluorescence. Recorded cells and post hoc visualized cells were matched by reference to the positions of the glass electrode tips relative to the brain surface. The locations of LFP electrodes and recorded cells were measured; the positions of the soma and the electrode tips were roughly aligned along the proximodistal axis in sections, and their coordinates were then three-dimensionally reconstructed and calculated on the x-(mediolateral) and y- (anteroposterior) axes relative to the bregma.

To confirm the expression of eNpHR3.0-EYFP in PV-positive interneurons (Supplementary Fig. 7), we conducted immunohistochemical staining after post hoc visualization of patch-clamped cells. For immunohistochemical staining, sections were blocked with 10% goat serum and 0.3% Triton X-100 in PBS for 60 min and incubated with a chicken primary antibody against green fluorescent protein (GFP; 1:1000, ab13970, Abcam) and a guinea pig primary antibody against parvalbumin (1:500, 195 004, Synaptic Systems) for 16 h. Sections were washed three times for 10 min with PBS and incubated with Alexa Fluor 488-conjugated goat secondary antibody against chicken IgG (1:500, A11039, Thermo Fisher Scientific), Alexa Fluor 594-conjugated goat secondary antibody against guinea pig IgG (1:500, A11076, Thermo Fisher Scientific), and blue fluorescent NeuroTrace (1:500, N21479, Thermo Fisher Scientific, MA, USA) for 6 h.

**ΔVm analysis.** Data were analyzed offline using custom-made MATLAB (R2017b, Natick, Massachusetts, USA) routines. The summarized data are reported as the means ± SDs unless otherwise specified. For box plots, the centerline shows the median, the box limits show the upper and lower quartiles, the whiskers cover 10–90% quantiles, and the points are all data points. For correlation plots, the significance was determined based on Pearson's correlation coefficient and a t-test of the correlation coefficients. $P < 0.05$ was considered statistically significant. All statistical tests were two-sided.

To detect SWRs from LFP traces recorded by a tungsten electrode, LFP traces were downsampled to 500 Hz and bandpass filtered between 100 and 250 Hz. Ripples, referred to here as SWRs, were detected at a threshold of $3 \times SD$ of the baseline noise[59]. The detected events were subsequently scrutinized by eye and manually rejected if the detection was erroneous. We used patching datasets that included at least 30 SWRs for further analyses.

Spikes were detected as peaks during periods with Vm greater than −20 mV in raw 20-kHz traces of whole-cell recordings. To average subthreshold Vm values, spikes were truncated as follows: (i) raw Vm traces were smoothed using the

moving average with a window of 1 ms, (ii) the minimum index at which the rate of change exceeded 4 V/s was detected as the leading edge of a spike, and (iii) traces were linearly interpolated from the leading edge until the time point that first showed a $Vm$ value below the edge value. When multiple edges, such as those of complex spikes, were detected within a period of 30 ms, these edges were individually interpolated from the previous edge until the next time point that first showed a $Vm$ value below the edge value. They were aligned to the SWR onset times and additively averaged across SWRs.

To quantify $Vm$ during SWRs, we searched for the highest peak value of $Vm$ between −20 ms and +120 ms relative to the SWR onset time. $\Delta Vm_{pre}$ was computed for each SWR event accompanied by depolarization. To quantify $\Delta Vm_{pre}$, we first averaged $Vm$ between −2000 ms and −1000 ms relative to the SWR onset time as the baseline $Vm$. We then identified the time point that gave the minimum $Vm$ between −50 and 0 ms relative to the SWR onset time (pre-SWR). $Vm$ was averaged between −25 ms and +25 ms relative to the time point, and the baseline $Vm$ was subtracted to obtain $\Delta Vm_{pre}$.

For SWR-triggered analysis of excitatory and inhibitory postsynaptic conductances (EPSGs and IPSGs), we took a postsynaptic current at a given time between −2000 ms and +400 ms relative to the SWR onset and subtracted the mean current between −2000 ms and −1000 ms. The current traces were averaged across SWRs. EPSGs and IPSGs were calculated as $I_{E/I}/(V_h − E_{rev, E/I})$, where $I_{E/I}$ is the excitatory/inhibitory current at a given time; $V_h$ is the holding potential of −70 mV and +10 mV for EPSGs and IPSGs, respectively[30]; and $E_{rev\ E/I}$ are the reversal potentials of 0 mV and −90 mV for EPSGs and IPSGs, respectively[60].

**Optogenetic manipulation with patch-clamp and LFP recordings.** For optogenetic manipulation (Supplementary Fig. 7), PV-Cre mice (Stock No. 017320, The Jackson Laboratory) were used. Three- to four-week-old mice anesthetized with isoflurane (Pfizer Inc., New York, NY) were placed in a stereotaxic frame. AAV-Ef1a-DIO-eNpHR 3.0-EYFP ($1.56 \times 10^{13}$ VG/ml, 300 nl; Addgene plasmid # 26966) was unilaterally injected into the dorsal hippocampal CA1 area (caudal −1.9 mm, lateral +1.6 mm from the bregma, and ventral 1.0 mm from the surface of the brain) at a rate of 100 nl/min using a syringe pump (KD Scientific Inc., Holliston, MA, USA) connected to glass pipettes (30-0034; Harvard Apparatus, Holliston, MA). After surgery, the mice were returned to their home cages. More than 2 weeks after the virus injection, the mice were prepared for in vivo whole-cell recordings as mentioned above. Simultaneous optical stimulation (50-ms duration, 3-s intervals, 5–8 mW) was delivered using a 561-nm laser (MGL-FN-561, Changchun New Industries Optoelectronics Technology, Changchun, China) through optical fibers in the glass pipettes held by a homemade pipette holder with two holes for the silver wire and the optical fiber.

**Unit recordings.** For chronic extracellular recordings, adult wild-type C57BL/6J mice from Charles River Laboratory (5 male, 1 female; 4–6-month old) were anesthetized with 1.5–2% isoflurane and implanted with silicon probes (Cambridge NeuroTech; ASSY-156-E-1) directed at the dorsal CA1 region (2 mm posterior and 1.7 mm lateral to bregma). Half of the animals had probes implanted in the left hemisphere, while the other half had probes implanted in the right hemisphere. All animals expressed ChR2-EYFP in a subset of CA1 pyramidal cells (pAAV-CaM-KIIa-hChR2(H134R)-EYFP; Addgene plasmid # 26969), transfected with in utero electroporation. Each probe was mounted on a custom-built microdrive and implanted at a 45° angle along the anteroposterior axis at a depth of ~0.7 mm. Craniotomies were sealed with a mix of dental wax and mineral oil, and a copper mesh cage was constructed to provide shielding. A stainless steel screw placed over the cerebellum served as the ground. Postoperatively, animals received a single intramuscular injection of 0.06 mg/kg buprenorphine (0.015 mg/ml), followed by additional doses as needed for the following 1–3 days. Following a 7-day recovery period, neural signals were recorded in the home cage while probes were advanced into the CA1 pyramidal layer, which was identified physiologically via sharp wave reversal. Neural data were amplified and digitized at 30 kHz using Intan amplifier boards (RHD2132/RHD2000 Evaluation System, Intan). All recordings (39 sessions ranging in duration from 113.7 min to 473.7 min; median duration = 317.4 min) were performed in the home cage while animals cycled between sleep and wake status.

**Unit isolation and classification.** Spikes were extracted and classified into putative single units using Kilosort[61]. Manual curation was performed in Phy software with the aid of custom-built plugins. Throughout the manual curation step, isolation quality was judged by inspecting cross-correlograms for incorrect splits of single units (i.e., autocorrelogram structure detectable in the cross-correlogram). Cells were classified as putative pyramidal cells and interneurons via CellExplorer (https://cellexplorer.org). Briefly, putative interneurons were identified via hard thresholds imposed on the waveform shape (trough to peak) and the auto-correlogram rise and decay time constants. A total of 2490 well-isolated putative pyramidal cells and 965 putative interneurons were identified in this way.

**SWR detection from silicon probe recordings.** Ripples, referred to here as SWRs, were detected as described in Tingley et al.[62]. In short, wideband signals were downsampled to 1250 Hz and bandpass filtered between 130 and 200 Hz using a

fourth-order Chebyshev filter, and the normalized squared signal was calculated. SWR peaks were detected by thresholding the normalized squared signal at 5 SDs above the mean, and the surrounding SWR start and stop times were identified as crossings of 2 SDs around this peak. SWR duration limits were set to be between 20 and 200 ms. An exclusion criterion was provided by designating a 'noise' channel (no detectable SWRs in the LFP), and events detected on this channel were interpreted as false-positives (e.g., electromyography artifacts).

**Unit analysis.** Interneurons that fired prior to SWRs were identified by inspecting their peri-event time histograms around the onsets of SWRs (2.5-ms bins). Overall, peak firing times with respect to SWR onset were skewed toward positive values (peak firing following SWR onset). For statistical analysis, the 75 interneurons with the most negative peak firing times were selected. All of these values fell 1.5 SDs below the mean time of peak firing.

To estimate the interneuron firing rate prior to the onset of a single SWR, the spike trains of interneurons were smoothed using a Gaussian kernel with a full width at half maximum of 15 ms. The average rate in a 30-ms window prior to SWR onset was used as an estimate of the pre-SWR rate. The distribution of pre-SWR rates was divided into quintiles, effectively grouping SWRs by the magnitude of prior interneuron firing (Q1 = high firing rate quintile; Q5 = low-firing rate quintile).

For each pyramidal cell, the latency to the first spike and the rank order were computed for each SWR. Rank order was defined as the normalized temporal firing position in the sequence of all pyramidal cells participating in the SWR. The average latency to the first spike was then computed within each quintile. The difference in spike latency averages was computed between Q1 and Q5 to estimate delays in pyramidal cell spike latency (longer latencies in Q1 than in Q5) that were concomitant with changes in pre-SWR interneuron rates. To quantify the significance of these differences, latencies to the first spike were shuffled across quintiles, and the spike latency averages between Q1 and Q5 were recomputed. This procedure was repeated 500 times to generate a null distribution. Observed spike latency changes outside the 95% confidence interval were considered significant.

**Sequence analysis.** To capture the statistics of sequential firing within SWRs, HMMs with Poisson emissions were fitted to binned spike count data[33]. For each SWR, spikes from $N$ putative pyramidal cells were binned into nonoverlapping 15 ms bins, resulting in a sequence of spike count vectors $y_{1:T}$ that was treated as a single observation under the model. At any given time point, the network was assumed to dwell in one of $M$ latent states $S_t \in \{1, ..., M\}$, and transitions between states were assumed to be first-order Markovian, i.e., $P(S_t|S_{t−1}, S_{t−2}) = P(S_t|S_{t−1})$. Within each state, the spike count of each neuron was treated as arising from a Poisson process. Given a training set $D$ consisting of $X$ SWRs, the model likelihood is as follows:

$$P(D, S|\Theta) = \prod_{i=1}^{X} P(y^{(i)}_{1:T_i}|\Theta, S^{(i)}_{1:T_i}) P(S^{(i)}_{1:T_i}) \qquad (1)$$

where $\Theta \in \{\pi, A, \Lambda\}$ are the model parameters, $\pi$ is a $1 \times M$ vector specifying the probability distribution over states at the start of each sequence, $A$ is an $M \times M$ transition probability matrix, and $\Lambda$ is an $N \times M$ matrix holding the expected spike counts of the $N$ neurons in each of the $M$ states. Parameters were estimated using the expectation maximization (EM) algorithm, and the log-likelihood of each SWR sequence was calculated using the forward-backward algorithm. Across all sessions, the hyperparameter $M$, representing the number of states, was set to 15, a value identified via twofold cross-validation.

To compare the statistical structure of sequences of SWRs preceded by different interneuron firings, HMMs were fitted separately to Q1 and Q5 SWRs (the fastest and slowest pre-SWR rates, respectively), and the log-likelihoods (normalized to the lengths of the sequences) were evaluated for each set of SWRs. More specifically, the HMM was fitted to a particular quintile of SWRs, $Q_{train}$ (e.g., Q1), making the remaining quintile the test set, $Q_{test}$ (e.g., Q5). The difference of normalized log-likelihoods $LL(Q_{test}) − LL(Q_{train})$ was used to estimate the dissimilarity of the sequences across SWR quintiles. A negative value indicated that the HMM more accurately captured sequences on which it was trained than sequences from the test set. Each interneuron exhibiting pre-SWR firing associated with a pyramidal cell spike delay ($n = 64$ interneurons) yielded a unique grouping of SWRs into quintiles, and for each such grouping, two differences in likelihood values were estimated (fit on Q1, test on Q5; fit on Q5, test on Q1), resulting in $n = 124$ values. As a control, SWRs were randomly reassigned to quintiles, and the differences in log-likelihoods under refitted HMMs were computed.

**Interneuron firing heterogeneity in SWRs.** In recording sessions where we identified interneurons whose pre-SWR firing was associated with delayed spiking of pyramidal cells, a generalized mixture model framework was adopted using a Python toolbox (https://pomegranate.readthedocs.io/en/latest/GeneralMixtureModel.html) to further quantify the heterogeneity of interneuron firing across SWRs. For each SWR, we obtained the spike count across $N$ interneurons, resulting in vector $y_{(1:N)}$. These data were assumed to arise from one of $M$ discrete latent states ("ensembles"), where the spike count of each interneuron followed a Poisson distribution. Given a training

set $D$ consisting of $X$ SWRs, the model likelihood was as follows:

$$P(D|\Theta) = \prod_{i=1}^{X} \sum_{j=1}^{M} \pi_j P(y^{(i)}_{1:N}|\lambda_j), \qquad (2)$$

where $\Theta \in \{\pi, \lambda\}$ are the model parameters; $\pi$ is a vector of $M$ ensemble probabilities (summing to 1), and $\lambda$ is an $N \times M$ matrix of expected spike counts, one for each of the $N$ interneurons in each of the $M$ ensembles. Parameters were estimated via the EM algorithm. In each session, $M$ was a free parameter estimated via 20-fold cross-validation[63,64]. For a set value of $M$, spanning a range from 2 to 30, the test data log-likelihood (LL) was averaged across the 20 "test" folds and expressed as a deviance ($-2 \times$ LL). Deviance tended to decrease as $M$ increased across sessions (Fig. 9d, e), which we captured analytically using a monotonically descending exponential function. The number of ensembles $M$ was chosen to be the value that maximized the curvature of this function, i.e., the value just before the curve plateaus. With an identified number of ensembles $M$, the goodness-of-fit was assessed against a surrogate dataset. Briefly, the same 20-fold cross-validation was performed, with data LL computed on each test fold. In addition, we computed the LL on a surrogate fold, in which spike counts were randomly permuted across test set SWRs for each interneuron, thereby breaking their co-firing statistics while preserving their average spike counts. The difference between these LLs was stored, and the procedure was repeated 50 times for each fold, resulting in $50 \times 20 = 1000$ LL differences per session. Positive LL differences indicate that the LL of the original data exceeds that of their shuffled surrogates. The distribution of LL differences in each session was tested for significance using a one-sample $t$-test.

**Optically induced high-frequency oscillations**. To explore the relationship between the firing rates of interneurons and the spike timing of pyramidal neurons in the context of an artificially imposed network pattern, we considered a subset of recording sessions that involved optogenetically induced high-frequency oscillations[10,23]. Blue light pulses were 125–145 ms in duration and delivered at intervals randomly chosen from a range between 1.3 and 1.7 s. In each pulse, light intensity was ramped up using a 5 Hz quarter sinusoid with a pulse-like offset. Light was delivered in blocks of 5, with increasing light intensity across pulses. In all sessions, all light intensities produced high-frequency oscillations. A total of 500–1000 pulses were delivered per session. Blue light was delivered via a 100 µm diameter optic fiber attached to the silicon probe using laser diodes driven by the open-source Cyclops LED driver[65] (https://github.com/jonnew/cyclops). Our analysis focused on interneuron-pyramidal cell pairs in which pre-SWR interneuron firing affected pyramidal spike timing in spontaneously occurring SWRs. The intersection of these pairs with sessions that included optogenetically induced high-frequency oscillations resulted in a total of 50 interneuron-pyramidal cell pairs across 8 sessions and 3 animals. Pairs in which the pyramidal neuron discharged fewer than 10 spikes across light pulses were discarded, resulting in $n = 38$ pairs in the final analysis.

**Reporting summary**. Further information on research design is available in the Nature Research Reporting Summary linked to this article.

## Data availability

The extracellular dataset generated for the current study will be made publicly available in the Buzsáki lab repository (https://buzsakilab.nyumc.org/datasets/). Any additional data that support the findings of this study are available from the corresponding author upon reasonable request. Source data are provided with this paper.

## Code availability

All code used in this study is available from the corresponding authors upon reasonable request.

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

## Acknowledgements
This work was supported by JST ERATO (JPMJER1801), the Institute for AI and Beyond of the University of Tokyo, and JSPS Grants-in-Aid for Scientific Research (18H05525).

## Author contributions
A.N., R.H. G.B., and Y.I. conceptualized the study; A.N. and R.H. performed the experiments and data analysis; S.M. made viruses (AAV-Ef1a-DIO-eNpHR 3.0-EYFP); Y.I. wrote the original draft; and A.N., R.H., S.M., G.B. and Y.I. reviewed and edited the final manuscript.

## Competing interests
The authors declare no competing interests.
