## [Peer Review File · Nature Communications]

Inhibition allocates spikes during hippocampal ripplesREVIEWER COMMENTS

Reviewer #1 (Remarks to the Author):

Noguchi et al investigate the mechanisms defining the timing of pyramidal cell spikes during ripples. They find that inhibition preceding the onset of ripples correlates with the timing of pyramidal cell discharge during the ripple, suggesting that timed inhibition plays an important role in setting ripple-active sequences. The combination of methods used is impressive, the findings are of outstanding relevance for the field, and the conclusion of the study are well justified by the data. I have only minor issues with the work, which should be addressed by the authors:

- 1) The inclusion criteria for accepted whole-cell recordings are quite strict. I wonder how many cells that could be identified electrophysiologically were excluded due to insufficient filling and whether those neurons show similar responses. This can obviously only be reliably done with those neurons recorded in current clamp, for which the regular firing properties have been measured.
- 2) In addition to the recordings under anesthesia, patch-clamp recordings from 15 cells in awake mice were included, which is an important experiment. However, recordings from both sleep deprived mice and mice that were exposed to an enriched environment are pooled. These are quite different behavioural conditions. How many of the 15 neurons belong to which group and are there difference between cells from these two groups and?
- 3) Figure 6c shows that spatially clustered neurons receive preceding inhibition that is more similar, suggesting spatial clustering of inhibitory inputs. This is in line with distance-dependent decrease in connection probability and strength of GABAergic synapses. However, it also gives rise to the prediction that closely spaced pyramidal cells should often display more similar ripple spike times compared to distant pyramidal cells, which could be assessed in the large single-unit data set.
- 4) Comparing spike latency for ripple events of high and low preceding interneuron rate as done in Fig. 7 is an excellent piece of evidence that the level of ongoing inhibition relates to ripple spike timing. However, I wonder whether it is possible to causally test the impact of pre-ripple interneuron activation on ripple spike times. For instance, the authors could use their previously demonstrated optogenetically-induced artificial ripples approach (Stark et al., *Neuron* 83:467-480, 2014). With ChR2 in pyramidal cells and an inhibitory opsin in interneurons, the timing of interneuron silencing could be precisely controlled to occur before the induction of ripples by ChR2 activation. This manipulation should result in earlier ripple spike times of pyramidal cells.

Reviewer #2 (Remarks to the Author):

Nogushi et al. report that inhibitory potentials of variable strength preceding sharp wave-ripple events (SWR) determine spike timing of CA1 pyramidal cells. Using whole-cell and simultaneous field potential recordings in anesthetized (and some awake) mice they show that spike latency during SWR correlates with the amplitude of the preceding IPSP. Moreover, multi-cellular recordings reveal that the order of firing (sequence) correlates with the relative strength of inhibition. Inhibitory strength varies between individual SWR events, leading the authors to conclude that 'early inhibitory activity determines the sequential spike times of pyramidal cells and diversifies the repertoire of sequence patterns.'

This work has a very high technical standard, and it adds an interesting new finding to the question of spike sequence generation which, in turn, is important for current concepts on memory formation and -consolidation. Using advanced in vivo recording methods, the authors make a convincing point for the correlation between inhibition and spike timing in this network state. Nevertheless, this reviewer has one conceptual issue concerning causality and a number of specific questions and concerns which might be addressed or clarified by the authors:

Conceptual issue:

The authors suggest a causal role for inhibition in determining sequential spike timing in PYR cell assemblies during SWR (e.g., last sentence of Abstract). While they do briefly mention that further factors are determining firing time (Discussion, lines 213-217), the concept is still not easy to reconcile with the widely accepted assumption that excitatory synaptic plasticity plays a dominant role in the formation of space-encoding assemblies. Given the highly divergent axonal plexus of interneurons and their high degree of synchronization, it is not clear how differential inhibition of selected pyramidal cells could be achieved in such a way that a given PYR fires at the different, highly specific time points on a given SWR. Therefore, the authors might reconcile whether they claim that the amplitude of DVm-pre correlates with spike timing or whether they really claim that differential inhibition on individual SWRs is a causal mechanism in the formation of assemblies/sequences. The former, weaker claim may be a consequence of variations in E/I-balance and excitability. The latter, stronger claim would at least need a hypothesis on how the high specificity of inhibition in regulating sequence and/or assembly formation comes about. This stronger hypothesis seems to be favored by the authors, see, e.g., line 156: These partially correlated dynamics suggest the presence of cell assemblies. (Note: The authors do show that blocking inhibition shifts spikes to earlier phases, albeit with moderate effect size [Fig. 3a]. This intervention does show some causal relation between inhibition and spike timing, but it does not exclude that the specificity in sequential firing comes from excitatory connections).

Specific concerns:

1. Firing probability: Most of the analysis is on spike timing. However, only 49/263 cells fired action potentials during SWR at all (Results, first paragraph). Was overall inhibition during SWR different between active and inactive cells? Within single cells: was firing probability (no spiking, 1 spike, 2 spikes... per SWR) also influenced by pre-SWR IPSP amplitude?
2. The authors do not find any difference in temporal delay depending on the distance between recorded cells and field electrode (Results, lines 92-95 and 131-133; Fig. S1). It would be interesting to discuss this finding with respect to known propagation patterns and –velocities of SWRs within the range of recording size (see, e.g., the propagation velocity publishes by one of the authors in Patel et al., J Neurosci, 2013).
3. The plot in Fig. 2c shows IPSG consistently above EPSG, except the last ~1/3 of the SWR. The right panel seems to show the EPSG/EPSP line exactly on the diagonal for the last bit, which is in apparent contrast to the left graph, where IPSG it should be below EPSG. Can you please clarify this? The respective sentence in the Results section is not very well understandable when looking at the graph (lines 108-110): 'We then plotted the time evolution of the mean conductances around SWR events in the EPSG-versus-IPSG space (Figure 2c right) and found that IPSGs were dominant before SWRs and thereafter became linearly balanced with EPSGs.'
4. Timing of spikes: Did the authors find entrainment of spikes by ripples? While this is not the main topic of the present article it would be rather easy to analyze and increase confidence that the precision of spike timing was not affected by recording conditions. The overall distribution of spike times during SWR (Fig. 1c) appears relatively broad. Could the authors comment on the distribution compared to similar data by N. Maier, L. Menendez de la Prida and others?

Reviewer #3 (Remarks to the Author):

The ms by Noguchi et al. addresses the spike timing of hippocampal CA1 pyramidal cells (PCs) during sharp-wave-ripple activity (SWR). By performing challenging in vivo multi-patch recordings from up to 4 PCs simultaneously in anaesthetized animals, they demonstrate that PCs are hyperpolarized transiently (~50 ms) prior to SWRs and that the level of hyperpolarization/depolarization (Vm-pre) correlates with the latency of PC discharge (or peak

depolarization when the cells do not discharge) during SWR. The amplitude of the hyperpolarization shows strong dynamics over space and time, resulting in variable spike timing for the individual cells. They further show that the pre-SWR hyperpolarization is partially removed by intracellular blockade of Cl⁻-mediated inhibition, pointing to GABA-A receptor involvement. Finally, discharge of GABAergic interneurons (INs) in the pre-SWR correlates with latency of PC discharge latency in multi-unit recordings. The authors conclude that GABA-A receptor-mediated inhibition produced by an unidentified subset of INs firing before the onset of SWR is a major factor setting the latency of individual PCs, thereby the sequence of the PC assembly during SWRs.

The study is technically excellent and the results are convincing. However, a general shortcoming of the paper is the a lack of quantification of electrophysiological parameters: e.g. there is no quantitative data on the amplitudes of preSWR hyperpolarization, spike latencies, conductances, etc in the text – only in graphical form in the figures can the reader find some information.

A technical issue concerns the use of intracellular cesium fluoride and DIDS. This combination has, indeed, been shown to partially block GABA-A receptor mediated transmission (whereby fluoride seems to be the major active component), however it has strong side effects and can also reduce excitatory transmission (Atherton et al., 2016 PLoS One). The results remain plausible, but the authors should address the lack of specificity in their experimental setting. In this context the question emerges if the IPSPs recorded in the experiments were also blocked by the application CsF-DIDS. Do the authors have relevant data?

The conclusions are consistent with the data, however, there is a certain discrepancy between the context of the study (flexible forward and reverse replay of spike sequences) and the finding of highly variable spatio-temporal inhibition. The proposal that different level of pre-SWR inhibition determines the spike timing is convincing (one could say, trivial), it is not clear how the highly “dynamic” spatio-temporal inhibition can define strict forward and reverse replay. In the Discussion the authors argue that the experimental paradigm of the study is not well suited to directly address this question. On the other hand, they also state that forward and reverse sequences of depolarisation were observed in the experiments - this, however, is not shown in the Results. If anything, the data in fig. 6 argues against this notion. This conclusion needs to be revised and the relevant aspects in the Discussion clarified and/or Results extended.

Minor points:

Lines 37, 102, 620: the sentences state that the pre-SWR hyperpolarization was “abolished”. The figures (Fig 2b,c) suggest that it was reduced but not abolished. See also major comment above.

Line 39/40: “Thus, early inhibitory activity” - what is “early”? Please reword.

Line 63: “mutually nonexclusive” – “not mutually exclusive” would be more appropriate formulation here in my mind.

Lines 85-87: Could you please provide number of recorded cells? Also, probably more relevant than the recording times (2097 s - how much is that in hrs:mins?) would be how many SWR events were recorded and analyzed.

Line 159: “VmpreS” - typo?

Line 361-365: The way Vm-pre is calculated suggests that we are not looking at the strength of inhibition alone, but the level of Vm which is set by the balance of slow depolarization/excitation and fast pre-SWR hyperpolarization/ inhibition. The narrative of the ms focuses on the importance of inhibition, but it is not clear how much of the observed variability of Vm-pre derives from the variability of excitation/the depolarizing ramp. Could you please provide relevant data?

Figure 2c (right): This plot is interesting but lacks the information on time/timing. Could you please add a time course plot of the dEPSP/dISPG ratio to the left panel?

Please provide a definition for pre-SWR, peri-SWR and post-SWR periods.

The paper would need a through check of wording, grammatical errors and formatting.

There are some errors and inconsistencies in the figure legends. E.g. in figure 4 legend title there is a typo "per-SWR"; in figure 7b, the labelling of the graphs is not consistent with the figure legend, with mentions of magnitude instead of firing rate and not addressing cumulative probability.

Reviewer #1

Noguchi et al investigate the mechanisms defining the timing of pyramidal cell spikes during ripples. They find that inhibition preceding the onset of ripples correlates with the timing of pyramidal cell discharge during the ripple, suggesting that timed inhibition plays an important role in setting ripple-active sequences. The combination of methods used is impressive, the findings are of outstanding relevance for the field, and the conclusion of the study are well justified by the data. I have only minor issues with the work, which should be addressed by the authors:

Thank you for the positive evaluations, which have encouraged us to resubmit this manuscript. We have revised our manuscript in accordance with your comments. Our point-by-point responses are as follows:

1-1) The inclusion criteria for accepted whole-cell recordings are quite strict. I wonder how many cells that could be identified electrophysiologically were excluded due to insufficient filling and whether those neurons show similar responses. This can obviously only be reliably done with those neurons recorded in current clamp, for which the regular firing properties have been measured.

Thank you for asking this question. Indeed, we did not mention this in the first version of the manuscript. Most of the electrophysiologically identified pyramidal cells could be successfully visualized *post hoc*. However, we excluded many recordings simply due to the number of SWR events. The conditions of mice were largely heterogeneous in terms of the frequency of SWRs. If a mouse emitted few SWRs, we were not able to perform statistically acceptable analyses. Thus, we set the criteria to more than 30 SWRs during the recording periods. These criteria reduced the number of recordings from 255 to 64, causing the dissociation between the numbers of recordings mentioned at the beginning of the Results section and the numbers of actually analyzed data points. We have added a statement about the numbers of SWR events in each recording as the inclusion criteria for the analyses (LL. 483-484).

1-2) In addition to the recordings under anesthesia, patch-clamp recordings from 15 cells in awake mice were included, which is an important experiment. However, recordings from both sleep deprived mice and mice that were exposed to an enriched environment are pooled. These are quite different behavioural conditions. How many of the 15 neurons belong to which group and are there difference between cells from these two groups and?

Thank you for noting this essential point. Of the 15 neurons, 6 neurons were from sleep-

deprived mice, and 9 neurons were from mice that were exposed to the enriched environment. Unfortunately, six neurons did not produce enough SWRs for data analysis. On the other hand, the data from anesthetized mice in the main part of this study were obtained exclusively from mice that experienced the enriched environment. To maintain consistency throughout the manuscript, we included only the data obtained from the mice that were exposed to the enriched environment (Figure 4d). This change produced a similar result. We also removed the description about sleep deprivation in the Methods section (LL. 419-420).

1-3) Figure 6c shows that spatially clustered neurons receive preceding inhibition that is more similar, suggesting spatial clustering of inhibitory inputs. This is in line with distance-dependent decrease in connection probability and strength of GABAergic synapses. However, it also gives rise to the prediction that closely spaced pyramidal cells should often display more similar ripple spike times compared to distant pyramidal cells, which could be assessed in the large single-unit data set.

Thank you for raising this important question. While the current view of the hippocampus is that the probability of place field representation is quite random for closely spaced and distantly spaced neurons (Redish et al. 2001), your prediction that spatially close pyramidal neurons may behave similarly seems to be reasonable based on our data. One possible reason for the inconsistency is that we analyzed the depolarization peak times rather than the spike times. The idea is that spatially clustered pyramidal cells show more similar dynamics than distant cells at the subthreshold level, but the relationship becomes obscure when we observe firings of neurons. We agree that we could assess your prediction by analyzing large single-unit datasets as you suggested. One of the authors (RH) is currently working on a separate project addressing this issue. In the meantime, we mentioned this possibility as a future direction in the Discussion section (LL. 352-358).

1-4) Comparing spike latency for ripple events of high and low preceding interneuron rate as done in Fig. 7 is an excellent piece of evidence that the level of ongoing inhibition relates to ripple spike timing. However, I wonder whether it is possible to causally test the impact of pre-ripple interneuron activation on ripple spike times. For instance, the authors could use their previously demonstrated optogenetically-induced artificial ripples approach (Stark et al., Neuron 83:467-480, 2014). With ChR2 in pyramidal cells and an inhibitory opsin in interneurons, the timing of interneuron silencing could be precisely controlled to occur before the induction of ripples by ChR2 activation. This manipulation should result in earlier ripple spike times of pyramidal cells.

Thank you very much for this suggestion. We strongly agree with this reviewer in that the suggested approach would provide some of the strongest support for the “causality” of our claim. However, conducting the experiments requires a large amount of preparation and will take a couple of years. Additionally, by analyzing the data obtained, we will generate enough knowledge to write up another paper. Instead, we have made full use of our patch-clamp technique and directly approached the causality between interneuron activity and pre-SWR hyperpolarization. Because it is difficult to separately control optogenetic stimulation (*e.g.*, ChR2) and inhibition (*e.g.*, halorhodopsin) in the same area using different lights (please see Klapoetke et al., Nat Meth 11: 338–346, 2014 Figure 1), we sought to inhibit the activity of PV-positive interneurons expressing eNpHR by light stimulation while simultaneously recording CA1 LFPs and the membrane potentials of CA1 pyramidal cells (Supplementary Fig. 7a,b). Because the occurrence of SWRs is sudden and cannot be predicted in advance, we were not able to target the appropriate timing of light illumination (*i.e.*, immediately before the SWRs) in this method. Instead, we repeated illumination at a regular interval, and we picked up the data in which the interneurons were inhibited prior to SWRs by chance. As a result, we observed that pre-SWR hyperpolarization was reduced by optogenetic inhibition of PV-positive interneurons (Supplementary Fig. 7c). In addition, we observed a negative correlation between the depolarization peak times and the timings of stimulation onset relative to the subsequent SWR onset ($R = -0.45$, $P = 0.012$, $n = 10$ cells from 7 mice). This result indicates that inhibiting the activity of PV-positive interneurons immediately before SWRs advances the depolarization times of pyramidal cells during SWRs. We speculate that these results at least partially support the causal relationship of the pre-SWR activity of PV-positive interneurons to spike or depolarization times of pyramidal cells during SWRs.

We further analyzed a subset of recording sessions that involved optogenetically induced high-frequency oscillations (Figure 8a; Stark et al., 2014; Stark et al., 2015). When we looked at the interneurons that were activated before SWRs and significantly delayed some pyramidal cells during SWRs in Figure 7, light stimulation-induced changes in the firing rates were highly variable across the interneurons (Figure 8b), presumably due to the sparse expression of ChR2 and the anatomical structure of the local circuitry. This variability indicates that we could artificially activate or suppress some of the interneurons that were activated before SWR onset in spontaneous conditions. We then took advantage of this artificially induced increase or decrease in the firing rates of the pre-SWR activated interneurons and examined whether the induced changes in the interneuron firing rates resulted in the corresponding changes in the spike times of the associated pyramidal cells. As a result, a significant relationship between the interneuronal firing rates and the spike latencies of pyramidal cells was even observed here (Figure 8c). We also found that in these interneuron-pyramidal cell pairs, the interneurons were activated earlier than the pyramidal cells (Figure 8d). These results further support the idea that variable interneuronal activity leads to heterogeneity in the subsequent spike latencies of pyramidal cells, which should be embedded in the circuits.

Although these experiments are not exactly what the reviewer intended, we believe that these results support the causal relationship between the variable interneuronal activity and the subsequent spike latencies of pyramidal cells embedded in the circuits. Thus, we have added the results as Supplementary Fig. 7 and Figure 8 and described them in the Results section (LL. 160-173, 251-273). We also described the

additional experimental and analytical procedure in the Methods section (LL. 448-449, 461-470, 509-521, 530, 643-658). Furthermore, because we cannot conclude the causality for the activity of specific subtypes of interneurons and the sequential activity patterns of CA1 pyramidal cells even with the additional data, we further included the suggested experimental approach by the reviewer as a future direction in the last paragraph of the Discussion section (LL. 389-391).

Reviewer #2

Nogushi et al. report that inhibitory potentials of variable strength preceding sharp wave-ripple events (SWR) determine spike timing of CA1 pyramidal cells. Using whole-cell and simultaneous field potential recordings in anesthetized (and some awake) mice they show that spike latency during SWR correlates with the amplitude of the preceding IPSP. Moreover, multi-cellular recordings reveal that the order of firing (sequence) correlates with the relative strength of inhibition. Inhibitory strength varies between individual SWR events, leading the authors to conclude that 'early inhibitory activity determines the sequential spike times of pyramidal cells and diversifies the repertoire of sequence patterns.'

This work has a very high technical standard, and it adds an interesting new finding to the question of spike sequence generation which, in turn, is important for current concepts on memory formation and -consolidation. Using advanced in vivo recording methods, the authors make a convincing point for the correlation between inhibition and spike timing in this network state. Nevertheless, this reviewer has one conceptual issue concerning causality and a number of specific questions and concerns which might be addressed or clarified by the authors:

We thank the reviewer for these constructive comments, which have greatly improved our work. Individual responses are provided below:

Conceptual issue:

The authors suggest a causal role for inhibition in determining sequential spike timing in PYR cell assemblies during SWR (e.g., last sentence of Abstract). While they do briefly mention that further factors are determining firing time (Discussion, lines 213-217), the concept is still not easy to reconcile with the widely accepted assumption that excitatory synaptic plasticity plays a dominant role in the formation of space-encoding assemblies. Given the highly divergent axonal plexus is interneurons and their high degree of synchronization, it is not clear how differential inhibition of selected pyramidal cells could be achieved in such a way

that a given PYR fires at the different, highly specific time points on a given SWR. Therefore, the authors might reconcile whether they claim that the amplitude of DVm-pre correlates with spike timing or whether they really claim that differential inhibition on individual SWRs is a causal mechanism in the formation of assemblies/sequences.

The former, weaker claim may be a consequence of variations in E/I-balance and excitability. The latter, stronger claim would at least need a hypothesis on how the high specificity of inhibition in regulating sequence and/or assembly formation comes about. This stronger hypothesis seems to be favored by the authors, see, e.g., line 156: These partially correlated dynamics suggest the presence of cell assemblies. (Note: The authors do show that blocking inhibition shifts spikes to earlier phases, albeit with moderate effect size [Fig. 3a]. This intervention does show some causal relation between inhibition and spike timing, but it does not exclude that the specificity in sequential firing comes from excitatory connections).

Thank you for pointing out this fundamental issue related to our work. We agree that the currently dominant view for fine modulations of spike timing assumes pyramidal-pyramidal neuronal interactions. Based on this assumption, it is also tacitly assumed that CA1 assemblies are simply inherited or driven from the upstream region, most often from CA3. There seem to be obvious issues with this assumption. First, in a novel environment, CA1 neurons remap quickly, whereas CA3 neurons take several days to change (Mankin et al., PNAS, 2012). Second, sequential activity in CA1 can be induced optogenetically (i.e., without the need from upstream drive) and, importantly, the artificially induced sequences correlate with the native sequences. Critical in the present context, interneurons are the main 'drivers' of this sequential correlation (Stark et al., PNAS, 2015). Third, when artificial place fields are induced optogenetically in CA1 (again without a need for upstream drive), interneurons are critically involved in shaping the new place fields, and the secondarily driven interneurons differentially become incorporated into SWRs (McKenzie, Huszar et al., Neuron, 2021). Previous work has also shown that interneurons can have place fields (Marshall et al., 2002) and show phase procession (Maurer et al., 2006; Geisler et al., 2007). These observations are explained by the highly nonuniform innervation of interneurons by their parental pyramidal cells. Such heterogeneity is obvious from the wide range of monosynaptic convergence of pyramidal cells on their shared interneurons. Plastic changes in some synaptic transmissions between pyramidal cells and interneurons, especially from interneurons to pyramidal cells (McKenzie, Huszar et al., 2021), among heterogeneous connectivities are hypothesized to be a mechanism by which interneurons regulate sequence/assembly formation with high specificity through learning.

As one of the answers to the heterogeneity of interneuronal influence, we additionally analyzed the data that support the above view (Figure 9, LL. 275-292, 617-641). While interneurons are highly synchronous during SWRs, as you point out (Figure

9a), there is further heterogeneity underlying this effect (Figure 9b). Specifically, there are subsets of SWRs in which interneurons are more/less synchronous than they are on average. To quantify this observation, we fit a statistical model (generalized mixture model with multivariate Poisson observations) to extract ensembles of interneuron firing in SWRs (Figure 9c-e). Across all relevant sessions that have interneuron firing prior to SWRs (as we analyzed in Figure 7), there is further heterogeneity in interneuron firing throughout SWRs. More than 2 interneuronal ensembles could be detected across all sessions (Figure 9f), which reflects more structure than could be expected from different average SWR-related firing rates (Figure 9g). The ensembles reflect co-fluctuations around the interneurons' average firing rates (Figure 9h), so it is not so notable that there are different subsets of interneurons firing in different SWRs, while it is more relevant that they fire reliably to different extent across ensembles. Based on these additional data, it is expected that a downstream pyramidal cell "feels" different amounts of inhibition across different SWRs.

Overall, interneuron firing prior to SWRs is important for the spike times of pyramidal cells, as demonstrated by a causal perturbation in the intracellular data. Our new results indicate that there is a possible downstream effect of pre-SWR inhibition, in that interneuron firing throughout SWRs would also be affected, which would affect their downstream pyramidal cells, and so on. While this chain of effects is just speculation, at least we can now argue that there is heterogeneous interneuron firing throughout SWRs, which surely supports our statement that interneurons diversify the repertoire of pyramidal cell assemblies/sequences in SWRs.

Although our view is still that the heterogeneous activity of interneurons contributes to regulating assembly/sequence formation based on the additional results, the previous studies and the hypothesized mechanism mentioned above, we acknowledge that we do not have data supporting the causality unless we complete the additional experiment suggested by Reviewer #1, comment #1-4, and that the contribution of interneurons to the sequence/assembly formation should be a part of the mechanisms in addition to the well-examined excitatory drive. Therefore, we revised the description in the manuscript to clarify the stance of our argument (LL. 359-366, 389-391).

Specific concerns:

2-1) Firing probability: Most of the analysis is on spike timing. However, only 49/263 cells fired action potentials during SWR at all (Results, first paragraph). Was overall inhibition during SWR different between active and inactive cells? Within single cells: was firing probability (no spiking, 1 spike, 2 spikes... per SWR) also influenced by pre-SWR IPSP amplitude?

Thank you for the constructive comments. We conducted an additional analysis and made a new supplementary figure (Supplementary Fig. 5). For the difference between active and inactive cells (cells with at least one spike and no spike during SWRs, respectively), we calculated and compared the mean and the standard deviation (SD) of the $\Delta V_{m_{pre}}$ values. The SDs were significantly higher in inactive cells than in active cells, while no significant difference was found between the means of the two groups (Supplementary Fig. 5b,c). Based on the result, we speculate that the presynaptic interneurons of the active cells come to show heterogeneous activity rather than that the net inhibition to each neuron determines which cell to be active. Within single cells, we calculated the $\Delta V_{m_{pre}}$ values for each number of spikes per SWR event and pooled the data from all 64 neurons by Z-scoring the values of each cell. As a result, the plot showed a significant positive correlation between the Z-scored $\Delta V_{m_{pre}}$ values and the number of spikes per SWR (Supplementary Fig. 5a), indicating that pre-SWR hyperpolarizations affect not only the spike times but also the firing probability within single neurons. These results and related discussions are described in the Results section (LL. 143-150).

2-2) The authors do not find any difference in temporal delay depending on the distance between recorded cells and field electrode (Results, lines 92-95 and 131-133; Fig. S1). It would be interesting to discuss this finding with respect to known propagation patterns and –velocities of SWRs within the range of recording size (see, e.g., the propagation velocity publishes by one of the authors in Patel et al., J Neurosci, 2013).

Thank you for the suggestion. According to Patel et al., J Neurosci, 2013 Figure 8, the velocities of SWRs along the septotemporal axis are 300-400 $\mu\text{m}/\text{msec}$. Because the range of recording sites in the present study was $\varphi < 800 \mu\text{m}$, the time lag of SWRs by the positions of LFP electrodes could be less than 2 msec, which might be one-order smaller, compared to the time scale in our analyses (i.e., tens of milliseconds relative to SWRs) to affect the result. We have mentioned SWR propagation in the Discussion section (LL. 327-332).

2-3) The plot in Fig. 2c shows IPSPG consistently above EPSPG, except the last $\sim 1/3$ of the SWR. The right panel seems to show the EPSPG/EPSPG line exactly on the diagonal for the last bit, which is in apparent contrast to the left graph, where IPSPG it should be below EPSPG. Can you please clarify this? The respective sentence in the Results section is not very well understandable when looking at the graph (lines 108-110): ‘We then plotted the time evolution of the mean conductances around SWR events in the EPSPG-versus-IPSPG space (Figure 2c right) and found that IPSPGs were dominant before SWRs and thereafter became linearly balanced with EPSPGs.’

We apologize for the lack of clarity in the data representation. The period shown in the right panel of Figure 2c corresponds to a part of the period shown in the left traces. We have now added the green bar showing the period to the left panel and changed the plot in the right panel to the corresponding color (Figure 2c).

2-4) Timing of spikes: Did the authors find entrainment of spikes by ripples? While this is not the main topic of the present article, it would be rather easy to analyze and increase confidence that the precision of spike timing was not affected by recording conditions. The overall distribution of spike times during SWR (Fig. 1c) appears relatively broad. Could the authors comment on the distribution compared to similar data by N. Maier, L. Menendez de la Prida and others?

Thank you for the important suggestion related to the confidence of our data. We added a similar analysis to quantify spike entrainment to ripples in awake animals (Stark et al., 2014, Figure 3A) and presented the data in Supplementary Fig. 1. As in a previous report, spikes were phase-locked to ripples (Supplementary Fig. 1a; $P = 6.2 \times 10^{-6}$, $Z = 12.0$, *Rayleigh* test, $n = 1093$ spikes from 50 cells). The cycle-by-cycle histogram of spike times also showed ripple entrainment, where the most action potentials were observed within the cycle just before the ripple peak, as in the previous report (Supplementary Fig. 1b). These results have been described in the Results section (LL. 94-95).

For the distribution of spike times during SWRs, we believe that our data are similar to previous *in vivo* studies by L. Menendez de la Prida's laboratory (Valero et al., 2017, Figure 2D left) and ours (Csicsvari et al., 2000, Figure 3A CA1). Although our data have lower firing rates than those reports because of anesthesia, it seems to be consistent that the firing rate started to rise tens of milliseconds before SWRs and became approximately 3 times higher around the SWR peak times, which corresponded to tens of milliseconds after SWR onset in our data (Figure 1c). Although the data by N. Maier (Maier et al., 2011, Figure 8F,H) might show a slightly sharper distribution than our data, similar data to that of N. Maier were obtained under *in vitro* conditions, where a loss of axon fibers may lead to less spontaneous neural activity and require higher synchrony for SWR induction.

Reviewer #3

The ms by Noguchi et al. addresses the spike timing of hippocampal CA1 pyramidal cells (PCs) during sharp-wave-ripple activity (SWR). By performing challenging *in vivo* multi-patch recordings from up to 4 PCs simultaneously in anaesthetized animals, they demonstrate that PCs are hyperpolarized transiently (~50 ms) prior to SWRs and that the level of hyperpolarization/depolarization (V_m -pre) correlates with the latency of PC discharge (or peak depolarization

when the cells do not discharge) during SWR. The amplitude of the hyperpolarization shows strong dynamics over space and time, resulting in variable spike timing for the individual cells. They further show that the pre-SWR hyperpolarization is partially removed by intracellular blockade of Cl⁻-mediated inhibition, pointing to GABA-A receptor involvement. Finally, discharge of GABAergic interneurons (INs) in the pre-SWR correlates with latency of PC discharge latency in multi-unit recordings. The authors conclude that GABA-A receptor-mediated inhibition produced by an unidentified subset of INs firing before the onset of SWR is a major factor setting the latency of individual PCs, thereby the sequence of the PC assembly during SWRs.

We appreciate that this reviewer found scientific value in our manuscript. Thanks to the comments, we were pleased to be able to revise and improve the manuscript. Individual responses are listed below:

3-1) The study is technically excellent and the results are convincing. However, a general shortcoming of the paper is the a lack of quantification of electrophysiological parameters: e.g. there is no quantitative data on the amplitudes of preSWR hyperpolarization, spike latencies, conductances, etc in the text –only in graphical form in the figures can the reader find some information.

We thank this reviewer for raising this issue and apologize for the lack of quantitative information. We have added the means \pm SDs for all parameters quantified in the Figures (LL. 96-97, 110-113, 119-121, 137, 141, 153, 155-156, 179, 203-204, 211-213). We have also added the quantitative data for Figure 2b, where we referred to the reduction of pre-SWR hyperpolarizations based on the appearance of the traces (Figure 2b inset, LL. 110-113).

3-2) A technical issue concerns the use of intracellular cesium fluoride and DIDS. This combination has, indeed, been shown to partially block GABA-A receptor mediated transmission (whereby fluoride seems to be the major active component), however it has strong side effects and can also reduce excitatory transmission (Atherton et al., 2016 PLoS One). The results remain plausible, but the authors should address the lack of specificity in their experimental setting. In this context the question emerges if the IPSPs recorded in the experiments were also blocked by the application CsF-DIDS. Do the authors have relevant data?

Thank you for raising this important point. To prove that the IPSPs were blocked by the application of CsF-DIDS, we additionally conducted in vivo voltage-clamp recordings with CsF-DIDS in the intracellular solution. The IPSPs were drastically reduced by CsF-

DIDS application compared with the control experiments (Supplementary Fig. 3), indicating the blockade of inhibitory currents by the application of CsF-DIDS. We have shown the results in a supplementary figure (Supplementary Fig. 3) and described the results in the Results section (LL. 115-116). Furthermore, we quantified and compared the $\Delta V_{m_{pre}}$ values before and after the application of CsF-DIDS (Figure 2b inset; $P = 3.1 \times 10^{-29}$, $t_{6317} = -11.3$, Student's *t test*, $n = 597$ and 1,348 SWRs for control and CsF-DIDS conditions, respectively). Larger positive values in CsF-DIDS conditions also support that CsF-DIDS application blocked IPSPs rather than EPSPs. This result was described in the Results section (LL. 110-115).

3-3) The conclusions are consistent with the data, however, there is a certain discrepancy between the context of the study (flexible forward and reverse replay of spike sequences) and the finding of highly variable spatio-temporal inhibition. The proposal that different level of pre-SWR inhibition determines the spike timing is convincing (one could say, trivial), it is not clear how the highly "dynamic" spatio-temporal inhibition can define strict forward and reverse replay. In the Discussion the authors argue that the experimental paradigm of the study is not well suited to directly address this question. On the other hand, they also state that forward and reverse sequences of depolarisation were observed in the experiments - this, however, is not shown in the Results. If anything, the data in fig. 6 argues against this notion. This conclusion needs to be revised and the relevant aspects in the Discussion clarified and/or Results extended.

We apologize for the lack of adequate description. First, we agree that we did not show data about forward and reverse "replay"; therefore, we removed the description about forward and reverse replay from the Discussion section (LL. 334-336). Instead, we examined the $\Delta V_{m_{pre}}$ values of three simultaneously recorded cells when the identical three neurons depolarized in the forward or reverse order, in which the medial-to-lateral direction was anatomically defined as the "forward" order. Specifically, we sorted each triplet cell in order from their medial to lateral positions, divided the order of depolarizations into two directions, i.e., from medial to lateral and from lateral to medial, and separately plotted the $\Delta V_{m_{pre}}$ values of these two cases (Figure 5c). As a result, the order of the $\Delta V_{m_{pre}}$ values was also reversed when the order of depolarization was reversed. Although the result may not be surprising based on our other results, we have added these data in Figure 5c and suggested that identical triplet cells could receive the reverse order of inhibition in association with the reverse order of depolarizations. The description of Figure 5c has been added to the Results and Discussion sections (LL. 192-200, 336-340). We have also added the mechanisms underlying forward and reverse replays to the Discussion section (LL. 340-343). According to this additional data, we changed Supplementary Fig. 8, in which the anatomical axis has been changed from the

anteroposterior axis to the mediolateral axis (Supplementary Fig. 8, LL. 181-182).

Minor points:

3-4) Lines 37, 102, 620: the sentences state that the pre-SWR hyperpolarization was “abolished”. The figures (Fig 2b,c) suggest that it was reduced but not abolished. See also major comment above.

We have changed the word “abolished” to “reduced”, which we confirmed by the additional quantification (Figure 2b inset, L. 39). We also addressed the effect of CsF-DIDS on EPSCs and IPSCs by additional experiments. The details are referred to in the response to the major comment above.

3-5) Line 39/40: “Thus, early inhibitory activity” - what is “early”? Please reword.

We apologize for the confusing wording. We have rephrased the word “early” as “pre-SWR” (L. 41).

3-6) Line 63: “mutually nonexclusive” – “not mutually exclusive” would be more appropriate formulation here in my mind.

Thank you for pointing this out. We have replaced “mutually nonexclusive” with “not mutually exclusive” (L. 65). These pointers are very helpful for us, as non-English speakers.

3-7) Lines 85-87: Could you please provide number of recorded cells? Also, probably more relevant than the recording times (2097 s - how much is that in hrs:mins?) would be how many SWR events were recorded and analyzed.

We have shown the total number of recorded cells and SWR events at the beginning of the Results section (LL. 89-91). We have also revised the expression of the recording times by using 'min'.

3-8) Line 159: “VmpreS” - typo?

We used “V_mpreS” as the plural form of “V_mpre” (L. 190, 210, 902, 903).

3-9) Line 361-365: The way V_m -pre is calculated suggests that we are not looking at the strength of inhibition alone, but the level of V_m which is set by the balance of slow depolarization/excitation and fast pre-SWR hyperpolarization/ inhibition. The narrative of the ms focuses on the importance of inhibition, but it is not clear how much of the observed variability of V_m -pre derives from the variability of excitation/the depolarizing ramp. Could you please provide relevant data?

Thank you for the essential comment. We agree that our $\Delta V_{m_{pre}}$ values reflected the balance of excitation and inhibition. We analyzed the variability in Δ EPSGs and Δ IPSGs during the pre-SWR periods (-50~0 ms from SWR onset) and found that IPSGs were more variable than EPSGs ($P = 9.4 \times 10^{-12}$, $F = 0.077$, F -test, $n = 34$ and 57 SWR events from 5 cells). The data are shown in Supplementary Fig. 4. Thus, we think that the observed variability in $\Delta V_{m_{pre}}$ is accounted for more by IPSGs than EPSGs. We have addressed this matter in the Results section (LL. 126-128).

3-10) Figure 2c (right): This plot is interesting but lacks the information on time/timing. Could you please add a time course plot of the dEPSG/dISPG ratio to the left panel?

Thank you for the suggestion. We calculated the ratio of the mean EPSG to the mean IPSP and added its time course below the traces in Figure 2c left.

3-11) Please provide a definition for pre-SWR, peri-SWR and post-SWR periods.

We have provided the definition for pre-SWR periods (-50~0 ms from SWR onset) in ΔV_m analysis more clearly in the Methods section (L. 498). We removed the word “peri-SWR” because we did not analyze the peri-SWR data in this manuscript (L. 130, 321, 857 (Figure 3 legend title), 872 (Figure 4 legend title), 891 (Figure 5 legend title)). We only used this “peri-SWR” period for detecting SWR-relevant spikes or depolarizations, the method of which is described in the Methods section (LL. 494-495). We did not define post-SWR periods because we only referred to post-SWR periods in the Discussion and did not quantify them. In the previous report we referred to in the Discussion (English et al., 2014), the post-SWR period was defined as the periods after 150 ms relative to SWR peaks, which corresponds to the periods after SWRs we detected.

3-12) The paper would need a thorough check of wording, grammatical errors and formatting.

There are some errors and inconsistencies in the figure legends. E.g. in figure 4 legend title there is a typo "per-SWR"; in figure 7b, the labelling of the graphs is not consistent with the figure legend, with mentions of magnitude instead of firing rate and not addressing cumulative probability.

We apologize for these errors and inconsistencies. We have changed "per-SWR" in Figure 4 legend title to "SWR-relevant" and "magnitude" in Figure 7b legend to "firing rate" (L. 925). We have mentioned the cumulative probability in the legend of Figure 7b (LL. 926-927). We have hired a review company to check our manuscript to thoroughly revise grammatical errors and formatting. We also revised Figure 7, in which 'PYR' and 'INT' were changed to 'pyramidal cell' and 'interneuron', respectively, and added the description about error bars to Figure 7g legend (L. 950).

Other revised points

In the process of filling in the reporting summary, we added the statement, "All the statistical tests were two-sided", the definition of box plot elements, and the description about the correlation coefficients we used at the beginning of ΔV_m analysis in the Methods section (LL. 474-479).

Thank you for your reconsideration of our manuscript.

REVIEWER COMMENTS

Reviewer #1 (Remarks to the Author):

The authors have nicely clarified the points I raised. In particular the new PVI-silencing experiments and the analysis of artificially induced high-frequency oscillations substantially strengthen the causal role of inhibition for setting spike timing during SWRs. I have only three additional minor points, which do not require an additional round of review.

1. The authors included additional voltage clamp recordings using CsF-DIDS-containing pipettes (new Supplementary Fig. 3). These data are very convincing; however, the experiment should be described in greater detail in the text of the supplementary figure. Were the neurons clamped at 10 mV as in their previous voltage clamp recordings? I presume the same concentration of CsF-DIDS as for current clamp recordings was used? This could be briefly mentioned in the legend for clarity. Moreover, there is a formatting issue with the second p-value in the figure legend (should be superscript).

2. The authors show a convincing example of the effect of PVI silencing on pre-SWR hyperpolarization in the new Supplementary Fig. 7. It would be highly informative to show summary data of that result as well (i.e., average magnitude of the remaining hyperpolarization considering all light stimulations during the period of the pre-SWR inhibition. Judging from Fig. S7d, several such events during the pre-SWR inhibition period should be available from the current data.

3. I understand from the authors' response that the included experiments with artificially induced SWR-like high-frequency oscillations are based on an analysis of previously published data sets. This could be more clearly stated in the methods section (line 649/650).

Reviewer #2 (Remarks to the Author):

The authors have responded thoughtfully and extensively to my comments (as, as far as I can see, also to the othreviewer's concerns). The manuscript has been changed accordingly.

I have no further queries or comments, and I congratulate you to this very nice work.

Reviewer #3 (Remarks to the Author):

The authors have addressed all points raised and substantially revised the manuscript. I have no more comments/concerns.

Comments and Answers

Reviewer #1

The authors have nicely clarified the points I raised. In particular the new PVI-silencing experiments and the analysis of artificially induced high-frequency oscillations substantially strengthen the causal role of inhibition for setting spike timing during SWRs. I have only three additional minor points, which do not require an additional round of review.

Thank you for the positive evaluations of our additional data and these further constructive comments. We have revised our manuscript in accordance with your comments. Our point-by-point responses are as follows:

1. The authors included additional voltage clamp recordings using CsF-DIDS-containing pipettes (new Supplementary Fig. 3). These data are very convincing; however, the experiment should be described in greater detail in the text of the supplementary figure. Were the neurons clamped at 10 mV as in their previous voltage clamp recordings? I presume the same concentration of CsF-DIDS as for current clamp recordings was used? This could be briefly mentioned in the legend for clarity. Moreover, there is a formatting issue with the second *p*-value in the figure legend (should be superscript).

We apologize for the lack of information regarding the additional experiments and the oversight. We have now added the description of the holding voltage and the intrapipette solution in the figure legend (LL. 1041-1043). We have also revised the formatting issue with the second *p*-value (L. 1044).

2. The authors show a convincing example of the effect of PVI silencing on pre-SWR hyperpolarization in the new Supplementary Fig. 7. It would be highly informative to show summary data of that result as well (i.e., average magnitude of the remaining hyperpolarization considering all light stimulations during the period of the pre-SWR inhibition. Judging from Fig. S7d, several such events during the pre-SWR inhibition period should be available from the current data.

Thank you for the constructive suggestion. We have plotted the relationship between $\Delta V_{m_{pre}}$ and the timings of stimulation onsets relative to the subsequent SWR onsets, which showed a significant positive correlation (Supplementary Fig. 7d; $R = 0.39$, $P = 0.049$, *t*-test of the correlation coefficient, $n = 26$ SWRs from 10 neurons). The result suggests that pre-SWR hyperpolarizations were reduced by optogenetically silencing the activity of PV-positive interneurons during pre-SWR periods. We have added the description of this result in the Results section (LL. 170-176).

3. I understand from the authors' response that the included experiments with

artificially induced SWR-like high-frequency oscillations are based on an analysis of previously published data sets. This could be more clearly stated in the methods section (line 649/650).

We apologize for the misleading description in our response. All the data in this study, including the experiments with optogenetically induced SWR-like high-frequency oscillations, has been newly collected for this paper (LL. 398-400).

Reviewer #2

The authors have responded thoughtfully and extensively to my comments (as, as far as I can see, also to the othreviewer's concerns). The manuscript has been changed accordingly.

I have no further queries or comments, and I congratulate you to this very nice work.

We are very pleased to know that our revision of the paper met the requirements of this reviewer. We would like to thank you very much for the valuable comments.

Reviewer #3

The authors have addressed all points raised and substantially revised the manuscript. I have no more comments/concerns.

Thank you very much for the positive evaluation for our revised manuscript. We appreciate your constructive comments, which have led to the great improvement of our work.

Other revised points:

1. One citation has been added to the paragraph about the heterogeneity of interneurons because we referred to the analysis in this paper (LL. 287, 758-759).

2. One author, who made the virus used in the PV-eNpHR experiments, was omitted from the list of authors in the previous revision of the manuscript and has been additionally included in the author list and the contribution section (LL. 4, 844-846, 1017).